# Bayesian Attention Mechanism: A Probabilistic Framework for Positional Encoding and Context Length Extrapolation

**Arthur S. Bianchessi**[1]    **Yasmin C. Aguirre**[1]    **Rodrigo C. Barros**[1,2]    **Lucas S. Kupssinskü**[1]

[1]MALTA – Machine Learning Theory and Applications Lab, PUCRS – Porto Alegre, Brazil
[2]Kunumi Institute, Brazil

`{arthur.bianchessi, yasmin.cardozo}@edu.pucrs.br`
`{rodrigo.barros, lucas.kupssinsku}@pucrs.br`
`rodrigo.barros@kunumi.com`

## Abstract

Transformer-based language models rely on positional encoding (PE) to handle token order and support context length extrapolation. However, existing PE methods lack theoretical clarity and rely on limited evaluation metrics to substantiate their extrapolation claims. We propose the Bayesian Attention Mechanism (BAM), a theoretical framework that formulates positional encoding as a prior within a probabilistic model. BAM unifies existing methods (e.g., NoPE and ALiBi) and motivates a new Generalized Gaussian positional prior that substantially improves long-context generalization. Empirically, BAM enables accurate information retrieval at $500\times$ the training context length, outperforming previous state-of-the-art context length generalization in long context retrieval accuracy while maintaining comparable perplexity and introducing minimal additional parameters.

## 1 Introduction

Transformer-based neural models currently dominate language modeling (LM) due to their superior ability to capture variable-length token dependencies via self-attention. Nevertheless, transformers inherently lack positional information, requiring the incorporation of *Positional Encoding* (PE). PE is vital, particularly for enabling LMs trained on shorter contexts to generalize to significantly longer sequences during inference—a desirable capability known as *context length extrapolation*. However, the precise impact of PE on extrapolation remains poorly understood (Kazemnejad et al., 2023).

Several PE methods have been proposed to facilitate context length extrapolation, including Sinusoidal embeddings (Vaswani, 2017), RoPE (Su et al., 2024), ALiBi (Press et al., 2022), and even the omission of positional encoding entirely (NoPE) (Kazemnejad et al., 2023). Despite reported empirical successes in extrapolation, two critical issues persist: (i) many existing PE techniques are empirically motivated with limited theoretical foundations, and thus their behavior is not well-understood (Liu et al., 2024; Huang et al., 2023); (ii) evaluation methods rely heavily on perplexity, which may inadequately reflect true extrapolation capability, as LMs can achieve low perplexity simply through localized attention patterns, as demonstrated in sliding-window evaluations (Huang et al., 2023).

To address these issues, we introduce the Bayesian Attention Mechanism (BAM), a theoretical framework that reframes self-attention as an expectation of values computed under a joint probabilistic model of *content* and *position* of tokens. Within BAM, PE naturally emerges as a prior distribution over token positions, clarifying the theoretical basis of existing techniques. Notably, we illustrate how NoPE and ALiBi correspond explicitly to Uniform and Laplace positional priors, respectively.

Leveraging this robust theoretical foundation, we propose a new positional encoding strategy utilizing a Generalized Gaussian prior. Our approach[1] introduces fewer than 1,000 additional parameters yet delivers substantially improved extrapolation performance, demonstrated clearly in retrieval-based

---

[1]Code available at `https://github.com/ArthurSBianchessi/BAM`

tasks and traditional perplexity evaluations. Thus, BAM serves both as a unified theoretical framework for analyzing PE schemes and as a practical method for enhancing long-context attention.

## 2 BAYESIAN ATTENTION MECHANISM

In this section, we motivate the perspective of framing attention as a Bayesian mechanism, supporting both token content and position information. We show that PE strategies can be seen as priors of a Bayesian attention mechanism, hereby called BAM.

### 2.1 BAYESIAN ATTENTION AND THE JOINT PROBABILITY DISTRIBUTION $p_{ij}$

**Definition 1.** For a fixed query vector $\mathbf{q}_i \in \mathcal{R}^{1 \times d}$ and key-value matrices $\mathbf{K}, \mathbf{V} \in \mathcal{R}^{L \times d}$, a Bayesian Attention Mechanism computes self-attention as an expectation over its values:

$$\text{self-attention}(\mathbf{q}_i, \mathbf{K}, \mathbf{V}) = \frac{\exp\big(\text{score}(\mathbf{q}_i, \mathbf{k}_j)\big)}{\Sigma_z \exp\big(\text{score}(\mathbf{q}_i, \mathbf{k}_z)\big)} \mathbf{V} = \Sigma_j p_{ij} \mathbf{v}_j = \mathbf{E}_{j|i}[\mathbf{V}]$$

**Definition 1** states that the attention mechanism can be expressed as an expectation over values of the $i^{th}$ query, where $p_{ij}$ is the probability of token $j \in [1, L]$ when attended by token $i$. This definition is consonant with the self-attention mechanism defined by Vaswani (2017) and with prior attempts to frame self-attention as an expectation over values (Singh & Buckley, 2023), where the scoring function is the scaled dot product and $\sigma\big(\text{score}(\mathbf{q}_i, \mathbf{K})\big)$ computes $p_{ij} \forall j$. The term $p_{ij}$ is usually called the *attention weight*, however we frame it as a joint probability over content and positions.

**Definition 2.** In Bayesian Attention, $p_{ij}$ is a joint probability over the *content* of token $j$ ($f_{\text{cont}}$) and its *position* relative to query $\mathbf{q}_i$ ($g_{\text{pos}}$).

$$p_{ij} = p(f_{\text{cont}}(\mathbf{q}_i, \mathbf{k}_j)|g_{\text{pos}}(i, j)) \times p(g_{\text{pos}}(i, j))$$

**Definition 2** allows us to interpret $p_{ij}$ as a joint probability distribution dependent on both token content and position. Together, **Definitions 1 and 2** frame positional encoding as a probability distribution over the positions of the tokens within the context. Note that this definition, particularly in $p(f_{\text{cont}}(\mathbf{q}_i, \mathbf{k}_j)|g_{\text{pos}}(i, j))$, encompasses a dependency of content on position. This dependency is trivially modeled by a scalar $Z$ when the scoring function is additive, as detailed below.

When framing PE as that probability distribution over tokens in a context, we can derive parametrized probability distributions that explain positional encoding strategies such as NoPE (Kazemnejad et al., 2023) and ALiBi (Press et al., 2022), and propose novel PE strategies with known behaviors.

**Theorem 1**. If the scoring function of the attention mechanism is additive, i.e., of the form $f_{\text{cont}}(\mathbf{q}_i, \mathbf{k}_j) + g_{\text{pos}}(i, j)$, then $p_{ij}$ is the product of the marginal probabilities over *content* and *position*, dependent on a normalizing scalar $Z$:r

*Proof.* By Definition 1, we have that self-attention$(\mathbf{q}_i, \mathbf{K}, \mathbf{V}) = \sigma\big(\text{score}(\mathbf{q}_i, \mathbf{K})\big)\mathbf{V} = \Sigma_j \mathbf{v}_j p_{ij} = \mathbf{E}_{j|i}[V]$. Following the assumption that the scoring function is $f_{\text{cont}}(\mathbf{q}_i, \mathbf{k}_j) + g_{\text{pos}}(i, j)$, we have:

$$
\begin{aligned}
p_{ij} &= \sigma(\text{score}(\mathbf{q}_i, \mathbf{k}_j)) \\
&= \frac{\exp\big(f_{\text{cont}}(\mathbf{q}_i, \mathbf{k}_j) + g_{\text{pos}}(i, j)\big)}{\Sigma_z\Big(\exp\big(f_{\text{cont}}(\mathbf{q}_i, \mathbf{k}_z) + g_{\text{pos}}(i, z)\big)\Big)} \\
&= \frac{\exp\big(f_{\text{cont}}(\mathbf{q}_i, \mathbf{k}_j)\big) \cdot \exp\big(g_{\text{pos}}(i, j)\big)}{\Sigma_z\Big(\exp\big(f_{\text{cont}}(\mathbf{q}_i, \mathbf{k}_z) + g_{\text{pos}}(i, z)\big)\Big)} \\
&= \frac{\exp\big(f_{\text{cont}}(\mathbf{q}_i, \mathbf{k}_j)\big) \cdot \exp\big(g_{\text{pos}}(i, j)\big)}{\Sigma_z\Big(\exp\big(f_{\text{cont}}(\mathbf{q}_i, \mathbf{k}_z) + g_{\text{pos}}(i, z)\big)\Big)} \cdot \underbrace{\frac{\Sigma_z\Big(\exp\big(f_{\text{cont}}(\mathbf{q}_i, \mathbf{k}_z)\big)\Big) \cdot \Sigma_z\Big(\exp\big(g_{\text{pos}}(i, z)\big)\Big)}{\Sigma_z\Big(\exp\big(f_{\text{cont}}(\mathbf{q}_i, \mathbf{k}_z)\big)\Big) \cdot \Sigma_z\Big(\exp\big(g_{\text{pos}}(i, z)\big)\Big)}}_{=1}
\end{aligned}
$$

$$= \boxed{\frac{\exp\left(f_{\text{cont}}(\mathbf{q}_i, \mathbf{k}_j)\right)}{\Sigma_z\left(\exp\left(f_{\text{cont}}(\mathbf{q}_i, \mathbf{k}_z)\right)\right)}} \cdot \boxed{\frac{\exp\left(g_{\text{pos}}(i,j)\right)}{\Sigma_z\left(\exp\left(g_{\text{pos}}(i,z)\right)\right)}} \cdot \boxed{\frac{\Sigma_z\left(\exp\left(f_{\text{cont}}(\mathbf{q}_i, \mathbf{k}_z)\right)\right) \cdot \Sigma_z\left(\exp\left(g_{\text{pos}}(i,z)\right)\right)}{\Sigma_z\left(\exp\left(f_{\text{cont}}(\mathbf{q}_i, \mathbf{k}_z) + g_{\text{pos}}(i,z)\right)\right)}}$$

$$= \boxed{p(f_{\text{cont}}(\mathbf{q}_i, \mathbf{k}_j))} \cdot \boxed{p(g_{\text{pos}}(i,j))} \cdot \boxed{\frac{1}{\Sigma_z\left(p(f_{\text{cont}}(\mathbf{q}_i, \mathbf{k}_z)) \cdot p(g_{\text{pos}}(i,z))\right)}}$$

$$= \frac{\boxed{p(f_{\text{cont}}(\mathbf{q}_i, \mathbf{k}_j))} \cdot \boxed{p(g_{\text{pos}}(i,j))}}{\boxed{Z}}$$

$\square$

The distributions governing token content and position are thus dependent by a normalizing scalar factor $Z$. We can further interpret $Z$ from different perspectives, shedding light on the relationship between token content and position in self-attention (see Appendix K).

This derivation shows us that adding positional information in the scoring function of the self-attention mechanism leads to a product of probabilities over *content* and *position*. We now show that existing PE methods can actually be described by parametrized probability distributions, and that by defining particular distributions we can force the model to explicitly attend to long context.

## 2.2 POSITIONAL ENCODING AS PRIORS TO BAM

With **Theorem 1**, we can frame positional encoding as priors to BAM. In particular, we present lemmas that derive NoPE (Kazemnejad et al., 2023) and ALiBi (Press et al., 2022) as specific prior distributions to Bayesian self-attention.

---

**Lemma 1.** The causal mask in decoder models is a special case of BAM prior where

$$\text{Causal Mask} \Rightarrow p(g_{\text{pos}}(i,j)) = \text{Uniform}(1, i, j) \text{ over a given context } \mathbf{x}_{1,\dots,L}$$

---

**Lemma 2.** ALiBi is a special case of BAM prior where the token position distribution comprises both Uniform and Laplace distributions.

$$\text{ALiBi} \Rightarrow p(g_{\text{pos}}(i,j)) = \text{Uniform}(1, i, j) \cdot \text{Laplace}\left(0, \frac{1}{m}, j - i\right) \text{ over a context } \mathbf{x}_{1,\dots,L}$$

---

**Lemma 3.** ALiBi becomes local attention as the relative length $|j - i|$ increases.

$$\text{If } p_{ij} = \text{softmax}\left(\mathbf{q}_i \mathbf{K}^\top + \mathbf{M}_{i\bullet} + \mathbf{A}_{i\bullet}\right) \text{ then } \lim_{|j-i| \to \infty} p_{ij} = 0$$

---

*Proofs.* See Appendix B.1, B.2, and B.3.

## 2.3 A PE STRATEGY WITH A GENERALIZED GAUSSIAN AS PRIOR

Now we change the distribution over positions to be a Generalized Gaussian Distribution (GGD) and show its advantages over the existing PE methods. We call this new PE method GGD-BAM.

Let $\mathbf{B} = [b_{ij}]_{L \times L}$ where $b_{ij} = -\left|\frac{j-i-\mu}{\alpha}\right|^\beta$, for $i = 1, \dots, L$ and $j = 1, \dots, L$ be a matrix of non-linear biases that are added in the scoring function of the self-attention mechanism. This makes $p(g_{\text{pos}}(i,j)) = \text{GGD}(\mu, \alpha, \beta, j - i)$, for $\beta > 0$ and $\alpha > 0$. Self-attention is thus computed as:

$$\text{softmax}\left(\mathbf{q}_i \mathbf{K}^\top + \mathbf{M}_{i\bullet} + \mathbf{B}_{i\bullet}\right).$$

When $\mu = 0$, $\beta = 1$, and $\alpha = \frac{1}{m}$, we have an instance of ALiBi, i.e., a Laplace prior.

**Lemma 4.** GGD-BAM becomes local attention as the relative length $|j - i|$ increases, for any $\beta > 0$ and $\alpha \geq 0$.

$$\text{If } p_{ij} = \text{softmax}\left(\mathbf{q}_i \mathbf{K}^\top + \mathbf{M}_{i\bullet} + \mathbf{B}_{i\bullet}\right) \text{ then } \lim_{|j-i| \to \infty} p_{ij} = 0$$

*Proof.* See Appendix B.4.

**Theorem 2.** GGD-BAM is necessarily capable of seeing more context length than ALiBi:

$$\frac{\text{GGD-BAM}}{\text{ALiBi}} = \frac{\mathbf{B}_{ij}}{\mathbf{A}_{ij}} = \frac{-\left|\frac{j-i-\mu}{\alpha}\right|^\beta}{-m|j-i|} < 1 \text{ for some } \mu \text{ and } \alpha, \text{ and for } \beta < 1$$

*Proof.* See Appendix B.5.

## 2.4 RELAXING THE REQUIREMENT FOR $\beta > 0$

The requirement for $\beta > 0$ comes from the definition of a GGD. We already showed that by making $0 < \beta < 1$, GGD-BAM can see context beyond ALiBi. Now, we relax this requirement and allow $\beta < 0$. This effectively increases the size of the tail of the distribution and makes GGD-BAM capable of ignoring local context and focusing on *arbitrarily-long context*.

**Theorem 3.** GGD-BAM ignores local context for any $\beta < 0$ and $\alpha \geq 0$.

$$\forall \beta < 0, \forall \alpha \geq 0, \text{ if } p_{ij} = \text{softmax}\left(\mathbf{q}_i \mathbf{K}^\top + \mathbf{M}_{i\bullet} + \mathbf{B}_{i\bullet}\right) \text{ then } \lim_{|j-i| \to 0} p_{ij} = 0,$$

**Theorem 4:** GGD-BAM takes into account arbitrarily long context for any $\beta < 0$ and $\alpha \geq 0$.

$$\forall \beta < 0, \forall \alpha \geq 0, \text{ If } p_{ij} = \text{softmax}\left(\mathbf{q}_i \mathbf{K}^\top + \mathbf{M}_{i\bullet} + \mathbf{B}_{i\bullet}\right) \text{ then } \lim_{|j-i| \to \infty} p_{ij} \neq 0,$$

*Proofs.* See Appendix B.6 and B.7.

## 2.5 INTUITIVE EXPLANATION OF BAM

To complement the formal derivations provided so far, we include here an intuitive visualization of the positional priors of BAM. Figure 1 illustrates the probability distribution $p(g_{\text{pos}}(i, j))$ that modulates attention over token positions $j$ for a fixed query position $i$. These curves reflect the prior belief of the model over the relevance of each position $j$ when computing the attention for token $i$, as per the decomposition $p_{ij} \propto p(f_{\text{cont}}(\mathbf{q}_i, \mathbf{k}_j)) \times p(g_{\text{pos}}(i, j))$.

**Uniform Prior.** This distribution corresponds to the absence of any positional inductive bias beyond the causal mask (NoPE). All positions within the causal window (i.e., $j \leq i$) are assigned equal probability, whereas all positions outside the causal window ($j > i$) are assigned probability zero.

Since Definition 2 states that $p_{ij} \propto p(f_{\text{cont}}(\mathbf{q}_i, \mathbf{k}_j)) \times p(g_{\text{pos}}(i, j))$, the $p_{ij}$ of each token outside the causal window is zero. As proved in Theorem 1, this case reduces BAM to the NoPE baseline. The gray-shaded region in Figure 1 represents the non-causal part of the sequence, i.e., positions $j > i$, which are masked out in auto-regressive settings.

**ALiBi as a Laplace Prior.** The ALiBi mechanism injects a linearly increasing bias into the attention logits, which corresponds to a Laplace prior over relative position $|j - i|$. The resulting prior has a sharp peak near the query token and rapidly decays over distance. This can be seen in the green curve of Figure 1, which emphasizes local context. As demonstrated in Theorem 2, this behavior limits the attention window, making ALiBi sensitive to short-term dependencies but less effective at capturing long-range interactions. As the relative distance between query and key increases, the probability tends to zero, making ALiBi a local attention mechanism.

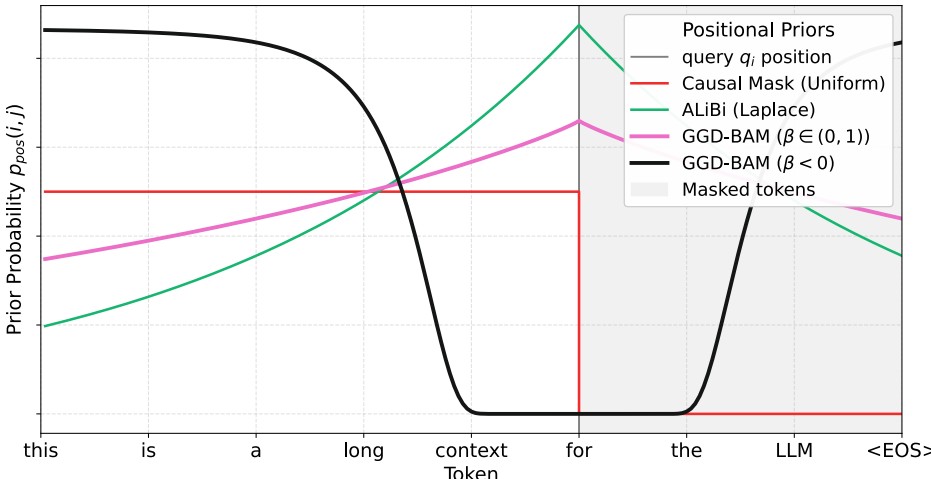

Figure 1: Visual comparison of different positional priors $p(g_{\text{pos}}(i, j))$ in BAM. Each curve represents the distribution over past token positions for a fixed query $\mathbf{q}_i$ in a fixed token position $i$.

**GGD-BAM with $\beta \in (0, 1)$.** The generalized Gaussian prior with fractional exponent produces a heavier-tailed distribution than Laplace. As shown in the pink curve, GGD-BAM maintains significant probability mass across distant tokens. This reflects the theoretical result in Theorem 2, where BAM with $\beta \in (0, 1)$ exhibits a slower decay rate and is able to attend to longer-range dependencies without diminishing the influence of distant content tokens as fast as ALiBi. The effect of the location parameter $\mu$ over the GGD is to move the peak along the distribution. If $\mu > 0$, it moves the peak beyond the causal mask. Among the parameters of the GGD, $\mu$ offers a smaller impact on the generalization of BAM, and could be fixed in $\mu = 0$ with little harm to perplexity and retrieval.

**GGD-BAM with $\beta < 0$.** Perhaps the most counterintuitive case is the use of an inverted generalized Gaussian prior. The black curve illustrates a scenario where the prior probability is effectively zero in the vicinity of the query position and sharply concentrated at the far end of the context window. Note that $\beta < 0$ is not a valid parametrization of the GGD in a strict probability sense. However, by relaxing the need for $\beta > 0$, we reach an intriguing theoretical result: the attention mechanism stops looking to local context and shifts to faraway tokens, allowing for arbitrarily-large context scenarios. Even though having attention heads with only negative $\beta$ would make the model blind to local context, attention heads with a negative exponent can act as retrieval heads capable of attending to very long context windows. This is corroborated by our results (see Section 3) where the longest passkey retrieval was achieved by a model in which some attention heads had a negative $\beta$.

Relaxing the requirement for $\beta > 0$ is important to increase context length extrapolation, though it is not desirable that all attention heads have $\beta < 0$. The parametrization $\beta < 0$ renders the model to be unable to attend to local context, and this increases its perplexity in language modeling. Therefore GGD-BAM is capable of not only encoding locality but also to explicitly suppress local content in favor of distant information. This behavior is beneficial in settings where important information appears in long-range context, such as causal reasoning and information retrieval.

**Interpretability and Control.** One of the central advantages of BAM is that these curves are not merely heuristics but correspond directly to explicit priors over token positions. This makes it possible to visualize, interpret, and even learn the attention pattern preferences of a model. Such a view also offers a principled mechanism for extrapolation beyond training context lengths, by selecting priors that maintain probability mass over long sequences.

**Scalable Softmax (SSMax).** Standard Softmax-based attention mechanisms suffer from a phenomenon known as *attention fading*, where the attention distribution becomes increasingly uniform as context length grows (Nakanishi, 2025). This occurs because the denominator in the Softmax computation increases with context size $n$, while the numerator for each token remains constant,

leading to vanishing attention peaks. To address this, Nakanishi (Nakanishi, 2025) proposes *Scalable Softmax* (SSMax), which rescales the attention logits dynamically as a function of sequence length: $z_i \mapsto \frac{n^{sz_i}}{\sum_{j=1}^n n^{sz_j}} = \frac{e^{(s \log n)z_i}}{\sum_{j=1}^n e^{(s \log n)z_j}}$, where $s \in \mathbb{R}$ is a learnable scalar. Although SSMax is not a PE method, it can be used to address a distinct and complementary challenge in long-context attention, the *fading* attention. This modification of the softmax function preserves the sharpness of the distribution regardless of input size, mitigating the flattening effect observed in softmax and improving long-context generalization.

While BAM models positional structure as an explicit prior over token positions, it becomes more susceptible to *fading attention* as it increases the context length that the LM can attend to. The two techniques are compatible and can be used together, enabling improved context length extrapolation both by a better normalizing factor via SSMax and in the distribution over positions using BAM. This learnable normalizing factor in the softmax function also serves the purpose of learning the normalizing dependence scalar $Z$ that we introduced during the derivation of BAM.

**The BAM matrix B.** To use GGD-BAM, the only modification we need to apply in the transformer is to add a relative position based matrix $\mathbf{B}$ to the attention score. Figure 2 shows how the attention score is computed for the entire query matrix $\mathbf{Q}$ using GGD-BAM. As usual, the causal mask $\mathbf{M}$ masks tokens beyond $\mathbf{q}_i$ by adding $-\infty$ to those respective positions. Each value in $\mathbf{B}$ is computed according to the relative position $j - i$, $\alpha$, $\beta$ and $\mu$. We fix $\mu = 0$ for most of our results.

Figure 2: Visual representation of the scoring function in GGD-BAM. The first matrix accounts for the content and the two others for the Uniform and GGD positional priors.

## 3 EMPIRICAL ANALYSIS

We perform an empirical analysis to evaluate the behavior of GGD-BAM in realistic long-context scenarios. We compare BAM and its variant coupled with Scaled Softmax (BAM SSMax) (Nakanishi, 2025) against several widely used PE methods: Sinusoidal (Vaswani, 2017), NoPE (Kazemnejad et al., 2023), RoPE (Su et al., 2024), Local RoPE (RoPE with sliding-window attention), and ALiBi (Press et al., 2022), as well as their versions coupled with Scaled Softmax.

All models presented in this section contain approximately 120M parameters, including ~25M for input embeddings and ~95M for the transformer layers. Models were trained on the FineWeb 10B dataset (Penedo et al., 2024) using the Mistral-7B v0.3 tokenizer (Jiang et al., 2023), with a training context length of 512 tokens. We evaluate model performance on two tasks: (1) language modeling on long-context samples drawn from FineWeb 10B and Wikipedia (Foundation, 2023); and (2) the Passkey Retrieval task (Mohtashami & Jaggi, 2023), which measures a model's ability to retrieve specific information from distant positions in the input sequence.

BAM introduces three learnable parameters, $\theta_\alpha$, $\theta_\beta$, and $\theta_\mu$, for each attention head in each layer. This results in a total overhead of $3 \times$ Heads $\times$ Layers parameters. In all experiments reported in this section, we train only $\theta_\alpha$ and $\theta_\beta$, fixing $\theta_\mu = 0$, which amounts to just 384 additional parameters in a 120M parameter model. Further implementation and training details are provided in Appendix C.

### 3.1 PASSKEY RETRIEVAL

To assess the capability of the LM to access and use long-range information, we evaluate the models on the Passkey Retrieval task (Mohtashami & Jaggi, 2023). This task measures whether a language model can recall a specific five digit number called the "passkey" embedded somewhere within a longer context window. In the passkey retrieval task, we generate 20 sequences, each containing a passkey inserted at uniformly spaced positions: $\frac{0L}{19}, \frac{1L}{19}, \ldots, L$. In the end of the sequence, we append the prompt `<The passkey is:>` and measure how accurate the model can predict the next five tokens. To avoid inaccuracies due to tokenization, each digit is considered a distinct token.

As shown in Figure 3, only models trained with BAM retained high accuracy when extrapolating to sequences up to 32,000 tokens. BAM SSMax maintained perfect accuracy across all tested lengths, demonstrating robust access to information throughout the full context window.

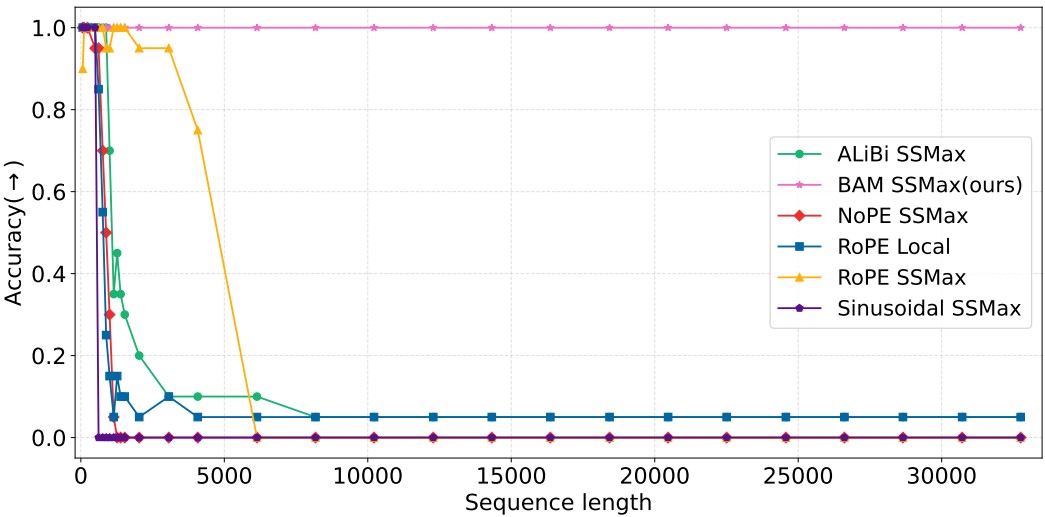

Figure 3: Passkey retrieval accuracy with distinct PE. BAM SSMax outperforms all PE methods maintaining perfect accuracy for a context beyond $64\times$ the training context length.

In contrast, all other evaluated PE methods such as Sinusoidal, RoPE, and NoPE rapidly degrade to near-random accuracy beyond their training horizon. Even ALiBi, which showed good perplexity extrapolation (see Appendix D), struggles to maintain retrieval performance at very long context windows. Appendix H.4 shows a similar trend to the evaluated PE methods without SSMax.

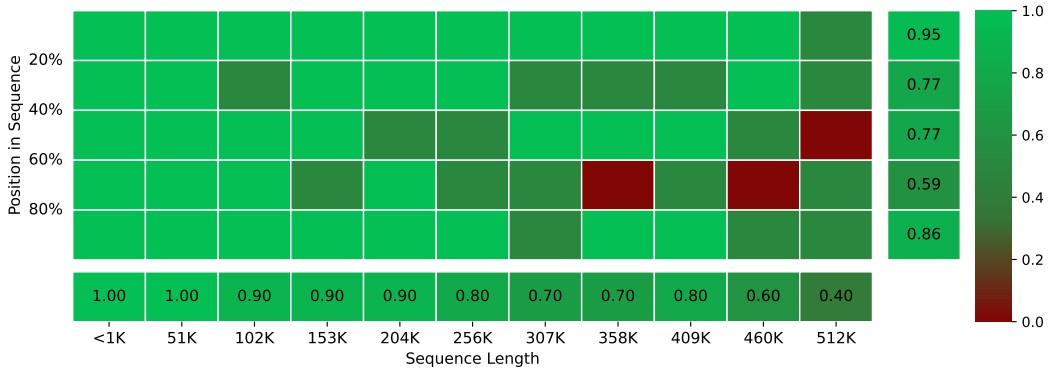

Figure 4: Passkey retrieval accuracy across context lengths and depths. The horizontal axis represent context length and the vertical axis represents the position of passkey in the context. In the bottom row and last column, we see average accuracy across length and position, respectively.

In Figure 4 we show a heatmap plot with passkey retrieval accuracy considering all possible depths of the passkey. We see that BAM is able to score perfectly in most lengths and depths while being trained only on length $512$. BAM only degrades to $0\%$ accuracy in $3/55$ of the evaluated lengths and depths. Although accuracy is above 80% for $500\times$ the training length, it appears that the model will degrade to zero eventually, however we did not have enough vram to test for longer context.

The superior performance of BAM in this retrieval setting provides empirical support that its positional prior enables meaningful access to distant content, rather than merely preserving surface-level fluency using local context. In Appendix D, we assess the perplexity of GGD-BAM in comparison to the baselines in Wikitext (Foundation, 2023) and Fineweb (Penedo et al., 2024). In Appendix F, we assess GGD-BAM in the Needle in a Haystack (NIAH) subset of the RULER benchmark (Hsieh et al., 2024), as a means to provide a thorough empirical analysis on long-context extrapolation.

## 3.2 Attention weights for $\beta \leq 0$, $\beta \in (0, 1)$ and $\beta \simeq 1$

To provide empirical evidence for the claims presented in Theorems 3 and 4, we visualize attention weights from individual attention heads and their respective values of $\beta$ in the Passkey Retrieval task.

To create this visualization, we craft a passkey retrieval prompt of length $841$ $(825 + 16)$, where the first 32 tokens are the *task prompt*–instruction to the model to remember the passkey; the following 25 tokens are the *passkey* itself; the next 768 tokens are *filler text* that the model should ignore; and the last 16 tokens are the retrieval *prompt answer* that the model should complete with the passkey.

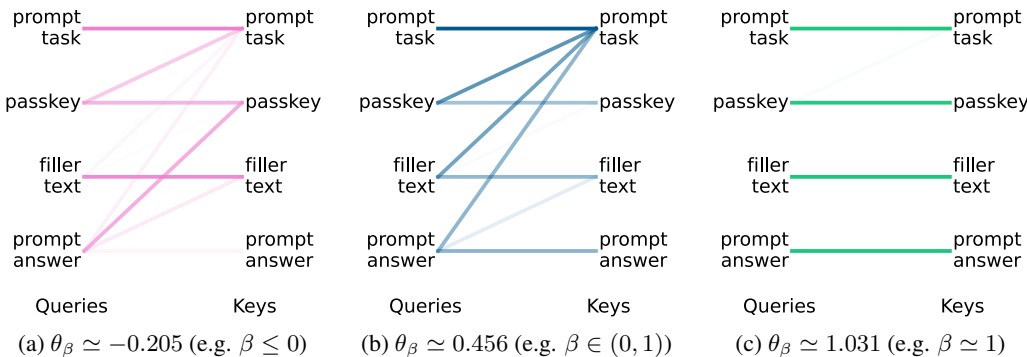

(a) $\theta_\beta \simeq -0.205$ (e.g. $\beta \leq 0$)   (b) $\theta_\beta \simeq 0.456$ (e.g. $\beta \in (0, 1)$)   (c) $\theta_\beta \simeq 1.031$ (e.g. $\beta \simeq 1$)

Figure 5: Attention weights from GGD-BAM during the Passkey Retrieval task. When $\beta \leq 0$, attention concentrates on distant keys (e.g., the passkey tokens), suppressing nearby content.

In Figure 5 (a), with $\beta \leq 0$, the attention head effectively ignores local context and sharply focuses on distant tokens—including the *passkey*—as predicted by our theoretical formulation. This behavior aligns with Theorems 3 and 4, which show that for $\beta \leq 0$, the GGD-BAM prior suppresses attention weights near the query token and sustains probability mass across long distances. As a result, these heads act as retrieval specialists, increasing the performance of the model for passkey retrieval.

In Figure 5 (b), with $\beta \in (0, 1)$, attention exhibits long-tail behavior, allocating attention mass more evenly across both nearby and distant positions. Compared to $\beta \simeq 1$, this head decays more slowly, retaining mid- and long-range dependencies. This supports our claim in Theorem 2 that GGD-BAM with fractional $\beta$ values can attend to longer contexts than ALiBi ($\beta \simeq 1$).

In Figure 5 (c), with $\beta \simeq 1$, the attention pattern is highly localized, closely resembling ALiBi's linear bias. Attention focuses on immediate neighbors, with a rapid decay over distance. This configuration is suitable for capturing local dependencies but is inadequate for long-range retrieval, as evidenced by its poor performance on the passkey task at large context lengths.

The visualizations in Figure 5 show that BAM is highly interpretable. The negative $\theta_\beta$ distribution that was conjectured to improve long-context retrieval appeared after training, and during inference it effectively caused higher attention weights towards the passkey that was further in the context.

### 3.3 $\theta_\beta$ AND $\theta_\alpha$ TRENDS AFTER TRAINING

We now analyze the trends of $\theta_\beta$ and $\theta_\alpha$ after training. Our first analysis focus on the $120M$ model trained with distinct context sizes. Figure 6 shows an interest trend where we identify three linearly-separable clusters of parameters: the first cluster with $\theta_\beta > 0$, where each attention head works similarly to a Laplace distribution (ALiBi); the second cluster with $-0.6 \leq \theta_\beta \leq 0$, which works as a retrieval head; the third cluster has fewer instances than the other two, with $\theta_\beta < -0.6$. We conjecture that this cluster works as a more aggressive retrieval head.

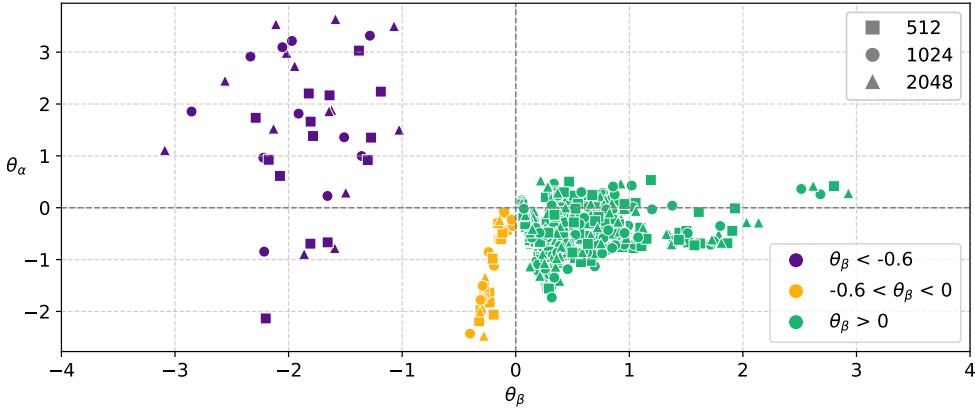

Figure 6: Trend of $\theta_\beta \times \theta_\alpha$ regarding the 120M model trained on $512$, $1024$, and $2048$ tokens.

When we analyze distinct model scales, the same three clusters emerge. Figure 7 shows the same clusters identified in Figure 6, providing evidence that these probability distribution over positions are stable and transferable across many tasks.

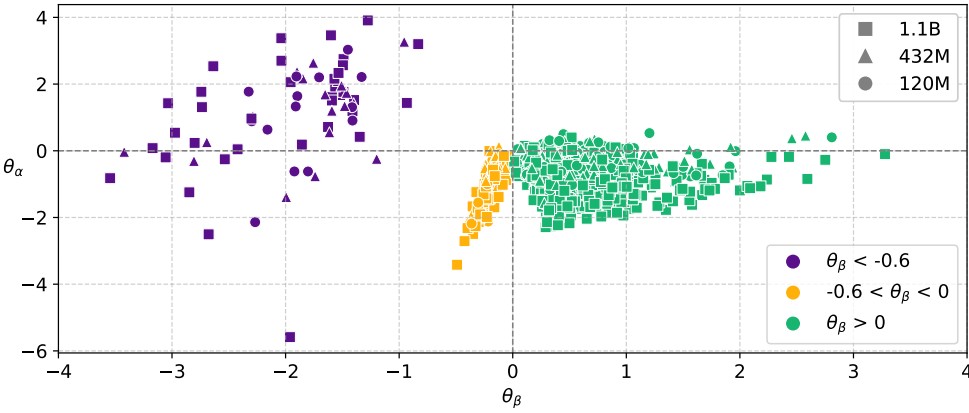

Figure 7: Trend of $\theta_\beta \times \theta_\alpha$ on different model scales.

In Appendices D, F, and H, we present additional empirical results for perplexity, for long-context performance on the Ruler Benchmark, as well as several additional ablation studies. We found that BAM has similar perplexity to ALiBi and has no measurable impact on inference time, while outperforming *every baseline* in context extrapolation across *all* evaluated tasks of the Ruler benchmark.

## 4 RELATED WORK

Several strategies for PE have been developed to allow Transformer-based LMs to encode token order. The most used PE methods are described below:

**Sinusoidal Positional Encoding.** Introduced by Vaswani et al. (Vaswani, 2017), sinusoidal encodings inject fixed positional information into the model by adding position-dependent vectors to token embeddings. In the self-attention mechanism, each $\mathbf{q}_i$, $\mathbf{k}_j$ and $\mathbf{v}_j$ have both content and positional information. BAM, instead, explicitly models position as a separate probabilistic component and does not add positional information directly into $\mathbf{q}_i$, $\mathbf{k}_j$, and $\mathbf{v}_j$.

**Rotary Position Embedding (RoPE).** RoPE (Su et al., 2024) encodes absolute positions through complex-valued rotations applied directly to $\mathbf{Q}$ and $\mathbf{K}$. RoPE encodes relative position by phase-shifting the token representations before the dot product. While RoPE introduces position at the dot-product level and preserves relative distance structure, it does not decouple content and position semantically. Han & Ji (2025) show that RoPE asymptotically disentangles semantics and position information in additive components in self-attention logits. Thus, we can see that RoPE asymptotically approximates BAM as the context length increases, though it lacks the flexibility and interpretability of assigning positional priors as established in our formulation of BAM.

**T5 Relative Bias.** T5 (Raffel et al., 2020) avoids absolute encodings entirely and instead learns a bias $b(i-j)$ for each relative distance $i-j$ that is added to the attention logits.This PE method could be viewed in BAM as an empirical non-parametric distribution over the positions $p(g_{\text{pos}}(i,j))$. However, both our theoretical grounding and experiments show that a parametric distribution with fewer trainable parameters can achieve superior context length extrapolation.

**No Positional Encoding (NoPE).** Haviv et al. (2022) and Kazemnejad et al. (2023) examined transformers without explicit PE. Although the attention mechanism is purely content-driven, the authors were the first to highlight that the causal mask $\mathbf{M}_{ij}$ (often not represented explicitly in the notation of other PE methods) in decoder-based transformers is sufficient to derive both absolute and relative PE.We showed NoPE to be a special case of BAM under a uniform positional prior.

**Attention with Linear Biases (ALiBi).** ALiBi (Press et al., 2022) injects a linear positional penalty into the attention scores. This formulation is interesting because it allows context length extrapolation in language modeling without introducing learnable parameters to the LM. We show that ALiBi is a special case of BAM with a Laplacian prior and we test it as an initialization strategy in our ablation study (see AppendixC). We explain how ALiBi maintains low perplexity in longer context windows and why it fails to retrieve information as it becomes local attention as the context length increases.

## 5 CONCLUSION

We introduced BAM, a principled probabilistic framework that reconceptualizes positional encoding as a prior over token positions within attention. By framing the attention mechanism as a factorized joint distribution over content and position, BAM not only offers a theoretical grounding for existing methods such as NoPE and ALiBi, but also motivates new families of positional priors. Our proposed Generalized Gaussian prior GGD-BAM significantly improves context length extrapolation in passkey retrieval task by more than $25\times$ compared to other PEs while maintaining low perplexity.

Despite its simplicity—increasing the parameter count of the model in negligible 0.00032% trainable parameters—GGD-BAM enables models to attend over significantly longer context windows without direct exposure during training. Experiments on FineWeb and Wikipedia show that BAM is uniquely able to recover distant information even at 250,000-token sequences, where other methods collapse. Moreover, our theoretical results demonstrate that BAM can express attention patterns that emphasize distant context or suppress locality, offering a new axis of inductive bias design for Transformers.

Future work includes applying BAM to larger models, further exploring the interpretability of learned positional priors, and extending the BAM framework to multi-modal input settings. Additionally, it remains an open question whether the extrapolation capabilities induced by BAM are preserved, or potentially enhanced, after instruction and preference fine-tuning. Investigating BAM under these downstream adaptation regimes is crucial for understanding its robustness in real-world applications. We further discuss all limitations of this work in Appendix I.

ACKNOWLEDGMENTS

This study was financed in part by the Coordination for the Improvement of Higher Education Personnel (CAPES) — Finance Code 001; by Conselho Nacional de Desenvolvimento Científico e Tecnológico (CNPq)— Grant Number: 443072/2024-8; and by Fundação de Amparo à Pesquisa do Estado do Rio Grande do Sul (FAPERGS) — Grant Number: 25/2551-0000891-3.

This work was supported by Kunumi Institute. The authors thank the institution for its financial support and commitment to advancing scientific research.

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

## A  PRELIMINARIES AND NOTATION

We denote by $\mathbf{x}_i$ the $i^{\text{th}}$ token in a sequence of length $L$, written as $\mathbf{x}_{1,\ldots,L}$. Each token is projected into *query*, *key*, and *value* via learned matrices: the query $\mathbf{q}_i \in \mathbb{R}^{1 \times d}$ is obtained from $\mathbf{x}_i \mathbf{W_q}$, and the full key and value matrices are given by $\mathbf{K} = \mathbf{X}\mathbf{W_k} \in \mathbb{R}^{L \times d}$ and $\mathbf{V} = \mathbf{X}\mathbf{W_v} \in \mathbb{R}^{L \times d}$, respectively.

We define $\mathbf{M}$ as the standard causal mask used to enforce auto-regressive decoding constraints. We use $\bullet$ to denote a slice from a matrix. For instance, $\mathbf{M}_{i\bullet}$ is the causal mask line vector for a single $\mathbf{q}_i$.

In our formulation, we interpret *attention weights* as a joint probability distribution $p_{ij}$ over two dependent components: *content* and *position*. The random variable $f_{\text{cont}}(\mathbf{q}_i, \mathbf{k}_j)$ denotes the content similarity between the $i^{\text{th}}$ query and the $j^{\text{th}}$ key, modeled by the conditional probability $p(f_{\text{cont}}(\mathbf{q}_i, \mathbf{k}_j))$. Similarly, the positional relation is modeled by the random variable $g_{\text{pos}}(i, j)$, which represents the relative position of token $j$ with respect to token $i$, captured by the prior distribution $p(g_{\text{pos}}(i, j))$.

It is a convention to denote the scoring function of the non-normalized logits in self-attention as $\text{score}(\mathbf{q}_i, \mathbf{K}) = \mathbf{q}_i \mathbf{K}^\top + \mathbf{M}_{i\bullet}$. To follow this convention, we adapt the notation used in ALiBi to present the linear biases in matrix form, so the scoring function becomes: $\text{score}(\mathbf{q}_i, \mathbf{K}) = \mathbf{q}_i \mathbf{K}^\top + \mathbf{M}_{i\bullet} + \mathbf{A}_{i\bullet}$. Finally, we introduce the BAM $\text{score}(\mathbf{q}_i, \mathbf{K}) = \mathbf{q}_i \mathbf{K}^\top + \mathbf{M}_{i\bullet} + \mathbf{B}_{i\bullet}$ by adding a matrix of non-linear biases to the scoring function.

Another convention we adopt is $\sigma(\mathbf{z}) = \text{softmax}(\mathbf{z}) = \frac{\exp(\mathbf{z}_i)}{\Sigma_j \exp(\mathbf{z}_j)}$.

**Positional Priors** The prior over positions, $p(g_{\text{pos}}(i,j))$, is modeled using parametric distributions over relative positions $|j - i|$. We consider the following parametric distributions:

- **Uniform distribution**: assigns equal probability mass to all valid positions before the current token:

$$p(g_{\text{pos}}(i,j)) = \text{Uniform}(a, b, x) \propto \mathbb{I}[a \leq x \leq b].$$

- **Laplace distribution**: decays exponentially with distance, controlled by a scale parameter $b > 0$:

$$p(g_{\text{pos}}(i,j)) = \text{Laplace}(\mu = 0, \alpha, x) \propto \exp\left(-\frac{|x|}{\alpha}\right).$$

- **Generalized Gaussian distribution (GGD)**: introduces a shape parameter $\beta > 0$ and scale $\alpha > 0$, allowing flexible control over decay behavior:

$$p(g_{\text{pos}}(i,j)) = \text{GGD}(\mu, \alpha, \beta, x) \propto \exp\left(-\left|\frac{x - \mu}{\alpha}\right|^{\beta}\right).$$

This formulation generalizes several priors: when $\beta = 1$ it recovers the Laplace distribution; when $\beta = 2$ it becomes Normal; larger values of $\beta$ yield sharper, more localized priors; lower values of $\beta$ yield large tailed distributions.

# B PROOFS

## B.1 NoPE IS A UNIFORM PRIOR TO BAM

**Lemma 1:** The causal mask in decoder models is a special case of BAM prior where

$$\text{Causal Mask} \Rightarrow p(g_{\text{pos}}(i,j)) = \text{Uniform}(1, i, j) \text{ over a given context } \mathbf{x}_{1,\dots,L}$$

*Proof.* The causal mask in a decoder model changes the scores of every token $\mathbf{x}_{i+1,i+2,\dots,L}$ to $-\inf$:

$$\text{causal-self-attention} = \text{softmax}(\mathbf{q}_i \mathbf{K}^\top + \mathbf{M}_{i\bullet})\mathbf{V}, \text{ where } \mathbf{M}_{i\bullet} = \begin{bmatrix} 0; & j \leq i \\ -\infty; & \text{otherwise} \end{bmatrix}.$$

Since $\mathbf{q}_i \mathbf{K}^\top$ has only *content* information and $\mathbf{M}_{i\bullet}$ has only *positional* information, we can use Theorem 1 to rewrite it as

$$\frac{\exp(\mathbf{q}_i \mathbf{k}_j^\top)}{\Sigma_z(\exp(\mathbf{q}_i \mathbf{k}_z^\top))} \cdot \frac{\exp(\mathbf{M}_{i\bullet})}{\Sigma_z(\exp(\mathbf{M}_{\mathbf{iz}}))} \cdot \frac{1}{Z}$$

Only the $\text{softmax}(\mathbf{M}_{i\bullet})$ term depends on the token position, so it is equivalent to $p(g_{\text{pos}}(i,j))$:

$$p(g_{\text{pos}}(i,j)) = \frac{\exp(\mathbf{M}_{i\bullet})}{\Sigma_z(\exp(\mathbf{M}_{\mathbf{iz}}))} = \begin{cases} \frac{1}{i}, & \text{if } j \leq i \\ 0, & \text{otherwise} \end{cases},$$

which is a Uniform distribution over tokens $\mathbf{x}_{1\dots i}$. $\square$

## B.2 ALiBi IS A PRIOR COMPRISING BOTH UNIFORM AND LAPLACE DISTRIBUTIONS

**Lemma 2:** ALiBi is a special case of BAM prior where the token position distribution comprises both Uniform and Laplace distributions.

$$\text{ALiBi} \Rightarrow p(g_{\text{pos}}(i,j)) = \text{Uniform}(1, i, j) \cdot \text{Laplace}\left(0, \frac{1}{m}, j - i\right) \text{ over a context } \mathbf{x}_{1,\dots,L}$$

*Proof.* Let $\mathbf{A} = [a_{ij}]_{L \times L}$ where $a_{ij} = -m|j - i|$, for $i = 1, \ldots, L$ and $j = 1, \ldots, L$ be the matrix with linear biases defined in ALiBi PE. The self-attention mechanism with ALiBi as PE is computed as:

$$\text{softmax}\left(\mathbf{q}_i \mathbf{K}^\top + \mathbf{M}_{i\bullet} + \mathbf{A}_{i\bullet}\right) \mathbf{V}$$

where $\mathbf{M}_{i\bullet}$ is the causal mask and $-m|j - i|$ are the linear biases added by ALiBi. According to Theorem 1 and Lemma 1, we can rewrite the softmax as:

$$\frac{\exp(\mathbf{q}_i \mathbf{k}_j^\top)}{\Sigma_z(\exp(\mathbf{q}_i \mathbf{k}_z^\top))} \cdot \frac{\exp(\mathbf{M}_{ij})}{\Sigma_z(\exp(\mathbf{M}_{\mathbf{iz}}))} \cdot \frac{\exp(\mathbf{A}_{ij})}{\Sigma_z(\exp(\mathbf{A}_{iz}))} \frac{1}{Z_{\text{ALiBi}}}$$

where $Z_{\text{ALiBi}}$ is a scaling (normalizing) factor. Lemma 1 further allows us to substitute the causal mask term with a uniform distribution:

$$\frac{\exp(\mathbf{q}_i \mathbf{k}_j^\top)}{\Sigma_z(\exp(\mathbf{q}_i \mathbf{k}_z^\top))} \cdot \text{Uniform}\,(1, i, j) \cdot \frac{\exp(\mathbf{A}_{ij})}{\Sigma_z(\exp(\mathbf{A}_{iz}))} \frac{1}{Z_{\text{ALiBi}}}$$

The softmax $(-m|j - i|)$ term has no content information, just positional, so we further work on it:

$$
\begin{aligned}
\frac{\exp(\mathbf{A}_{ij})}{\Sigma_z(\exp(\mathbf{A}_{iz}))} &= \frac{\exp\left(-m|j - i|\right)}{\Sigma_z \exp\left(-m|z - i|\right)} \\
&= \frac{\frac{m}{2} \exp\left(-m|j - i|\right)}{\Sigma_z \frac{m}{2} \exp\left(-m|z - i|\right)} \\
&= \frac{\text{Laplace}\left(0, \frac{1}{m}, j - i\right)}{\Sigma_z \frac{m}{2} \exp\left(-m|z - i|\right)} \\
&= \text{Laplace}\left(0, \frac{1}{m}, j - i\right)
\end{aligned}
$$

We can drop the scalar normalizing denominator $\Sigma_z \frac{m}{2} \exp\left(-m|z - i|\right)$ since $Z_{\text{ALiBi}}$ accounts for the normalization of the whole expression. Back to Definition 2, we have $p(g_{\text{pos}}(i, j)) = \text{Uniform}\,(1, i, j) \cdot \text{Laplace}\left(0, \frac{1}{m}, j - i\right)$. $\qquad \square$

### B.3 ALiBi is Local Attention for large $|j - i|$ lengths

> **Lemma 3:** ALiBi becomes local attention as the relative length $|j - i|$ increases.
>
> $$\text{If } p_{ij} = \text{softmax}\left(\mathbf{q}_i \mathbf{K}^\top + \mathbf{M}_{i\bullet} + \mathbf{A}_{i\bullet}\right) \text{ then } \lim_{|j-i| \to \infty} p_{ij} = 0$$

*Proof.* We prove this lemma for a fixed query $\mathbf{q}_i$. The scoring function of ALiBi has three components: $\mathbf{q}_i \mathbf{K}^\top$, $\mathbf{M}_{i\bullet}$, and $\mathbf{A}_{i\bullet}$. Let us take the limit of the scoring function:

$$\lim_{|j-i| \to \infty} \left(\mathbf{q}_i \mathbf{K}^\top + \mathbf{M}_{i\bullet} + \mathbf{A}_{i\bullet}\right) = \mathbf{q}_i \mathbf{K}^\top + \mathbf{M}_{i\bullet} + \lim_{|j-i| \to \infty} \left(\mathbf{A}_{i\bullet}\right)$$

We drop the limit in the causal mask $\mathbf{M}_{i\bullet}$ as the only effect of increasing the context size and the distance between $i$ and $j$ in the causal mask is making the mask bigger in size, but it stills follows the same formation law with 0 to the left of the query and $-\infty$ elsewhere.

$$\lim_{|j-i| \to \infty} \left(\mathbf{A}_{i\bullet}\right) = \lim_{|j-i| \to \infty} \left(-m|j - i|\right) = -\infty$$

When we plug $-\infty$ back into the scoring function we see that it becomes $-\infty$, and consequently the softmax becomes 0.

$$\lim_{|j-i|\to\infty} p_{ij} = \lim_{|j-i|\to\infty} \text{softmax}\left(\mathbf{q}_i\mathbf{K}^\top + \mathbf{M}_{i\bullet} + \mathbf{A}_{i\bullet}\right) = 0$$

□

## B.4 GGD-BAM IS LOCAL ATTENTION FOR LARGE $|j-i|$ LENGTHS

**Lemma 4:** GGD-BAM becomes local attention as the relative length $|j-i|$ increases, for any $\beta > 0$ and $\alpha > 0$.

$$\text{If } p_{ij} = \text{softmax}\left(\mathbf{q}_i\mathbf{K}^\top + \mathbf{M}_{i\bullet} + \mathbf{B}_{i\bullet}\right) \text{ then } \lim_{|j-i|\to\infty} p_{ij} = 0$$

*Proof.* This proof is similar to Lemma 3. We prove this lemma for a fixed query $\mathbf{q}_i$. Let $\mu = 0$. The scoring function of GGD-BAM has three components $\mathbf{q}_i\mathbf{K}^\top$, $\mathbf{M}_{i\bullet}$, and $\mathbf{B}_{i\bullet}$, lets take the limit of the scoring function and see how it behaves:

$$\lim_{|j-i|\to\infty} \left(\mathbf{q}_i\mathbf{K}^\top + \mathbf{M}_{i\bullet} + \mathbf{B}_{i\bullet}\right) = \mathbf{q}_i\mathbf{K}^\top + \mathbf{M}_{i\bullet} + \lim_{|j-i|\to\infty} \left(\mathbf{B}_{i\bullet}\right)$$

We drop the limit in the causal mask $\mathbf{M}_{i\bullet}$ as the only effect of increasing the context size and the distance between $i$ and $j$ in the causal mask is making the mask bigger in size, but it stills follows the same formation law with 0 to the left of the query and $-\infty$ elsewhere.

$$\lim_{|j-i|\to\infty} \left(\mathbf{B}_{i\bullet}\right) = \lim_{|j-i|\to\infty} \left(-\left|\frac{j-i}{\alpha}\right|^\beta\right) = -\infty$$

When we plug $\infty$ back into the scoring function we see that it becomes $-\infty$, and consequently the softmax becomes 0.

$$\lim_{|j-i|\to\infty} p_{ij} = \lim_{|j-i|\to\infty} \text{softmax}\left(\mathbf{q}_i\mathbf{K}^\top + \mathbf{M}_{i\bullet} + \mathbf{B}_{i\bullet}\right) = 0$$

□

## B.5 GGD-BAM SEES MORE CONTEXT THAN ALIBI

**Theorem 2:** GGD-BAM is able to see more context length than ALiBi

$$\frac{\text{GGD-BAM}}{\text{ALiBi}} = \frac{\mathbf{B}_{ij}}{\mathbf{A}_{ij}} = \frac{-\left|\frac{j-i-\mu}{\alpha}\right|^\beta}{-m|j-i|} < 1 \text{ for some } \mu \text{ and } \alpha, \text{ and for } \beta < 1$$

*Proof.* For GGD-BAM to see more context than ALiBi, it must be the case that the ratio between $\mathbf{B}_{ij}$ and $\mathbf{A}_{ij}$ $\frac{-\left|\frac{j-i-\mu}{\alpha}\right|^\beta}{-m|j-i|}$ is less than 1 for some $\beta < 1$ and for some $\alpha$ and $\mu$. Let $\mu = 0$, $\alpha = \frac{1}{\sqrt[\beta]{m}}$ and $\beta \in (0,1)$, then we have:

$$\frac{\mathbf{B}_{ij}}{\mathbf{A}_{ij}} = \frac{-\left|\frac{j-i}{\alpha}\right|^\beta}{-m|j-i|}$$

$$= \frac{-m|j-i|^\beta}{-m|j-i|}$$

$$= |j-i|^{\beta-1}$$

$$< 1$$

Since the ratio between BAM and ALiBi can be less then 1, BAM shrinks at a slower rate than ALiBi, effectively capturing longer contexts as $|j - i|$ increases. □

## B.6 GGD-BAM IGNORES LOCAL CONTEXT FOR $\beta < 0$ AND $\alpha > 0$

> **Theorem 3:** GGD-BAM ignores local context for any $\beta < 0$ and $\alpha > 0$.
>
> $$\forall \beta < 0, \forall \alpha > 0, \text{ If } p_{ij} = \text{softmax}\left(\mathbf{q}_i \mathbf{K}^\top + \mathbf{M}_{i\bullet} + \mathbf{B}_{i\bullet}\right) \text{ then } \lim_{|j-i|\to 0} p_{ij} = 0,$$

*Proof.* This proof is similar to Lemmas 3 and 4. We prove this lemma for a fixed query $\mathbf{q}_i$. Since $i$ is fixed for a query, $|j - i| \to 0$ implies that the token $j$ and $i$ are the same token. The scoring function of GGD-BAM has three components $\mathbf{q}_i \mathbf{K}^\top$, $\mathbf{M}_{i\bullet}$, and $\mathbf{B}_{i\bullet}$, lets take the limit of the scoring function and see how it behaves:

$$\lim_{|j-i|\to 0} \left(\mathbf{q}_i \mathbf{K}^\top + \mathbf{M}_{i\bullet} + \mathbf{B}_{i\bullet}\right) = \mathbf{q}_i \mathbf{K}^\top + \mathbf{M}_{i\bullet} + \lim_{|j-i|\to 0} \left(\mathbf{B}_{i\bullet}\right)$$

We drop the limit in the causal mask $\mathbf{M}_{i\bullet}$ as the only effect of $i$ and $j$ being the same token is that $\mathbf{M}_{ij} = 0$.

When the shape parameter $\beta$ is negative, then we can perform the following manipulation.

$$\lim_{|j-i|\to 0} \left(\mathbf{B}_{i\bullet}\right) = \lim_{|j-i|\to 0} \left(-\left|\frac{j-i}{\alpha}\right|^\beta\right) = \lim_{|j-i|\to 0} \left(-\left|\frac{\alpha}{j-i}\right|^{|\beta|}\right) = -\infty$$

When we plug $-\infty$ back into the scoring function we see that it becomes $-\infty$, and consequently the softmax becomes 0.

$$\lim_{|j-i|\to 0} p_{ij} = \lim_{|j-i|\to 0} \text{softmax}\left(\mathbf{q}_i \mathbf{K}^\top + \mathbf{M}_{i\bullet} + \mathbf{B}_{i\bullet}\right) = 0$$

Since the result of the softmax is zero, GGD-BAM does not attend to local context when $\beta < 0$ and $\alpha > 0$. □

## B.7 GGD-BAM CAN HAVE ARBITRARILY LONG CONTEXT FOR $\beta < 0$ AND $\alpha > 0$

> **Theorem 4:** GGD-BAM takes into account arbitrarily long context for any $\beta < 0$ and $\alpha > 0$.
>
> $$\forall \beta < 0, \forall \alpha > 0, \text{ If } p_{ij} = \text{softmax}\left(\mathbf{q}_i \mathbf{K}^\top + \mathbf{M}_{i\bullet} + \mathbf{B}_{i\bullet}\right) \text{ then } \lim_{|j-i|\to\infty} p_{ij} \neq 0,$$

*Proof.* This proof is similar to Lemmas 3 and 4 and Theorem 3. We prove this lemma for a fixed query $\mathbf{q}_i$. The scoring function of GGD-BAM has three components $\mathbf{q}_i \mathbf{K}^\top$, $\mathbf{M}_{i\bullet}$, and $\mathbf{B}_{i\bullet}$, lets take the limit of the scoring function and see how it behaves:

$$\lim_{|j-i|\to\infty} \left(\mathbf{q}_i \mathbf{K}^\top + \mathbf{M}_{i\bullet} + \mathbf{B}_{i\bullet}\right) = \mathbf{q}_i \mathbf{K}^\top + \mathbf{M}_{i\bullet} + \lim_{|j-i|\to\infty} \left(\mathbf{B}_{i\bullet}\right)$$

We drop the limit in the causal mask $\mathbf{M}_{i\bullet}$ as the only effect of increasing the context size and the distance between $i$ and $j$ in the causal mask is making the mask bigger in size, but it stills follows the same formation law with 0 to the left of the query and $-\infty$ elsewhere.

When the shape parameter $\beta$ is negative, then we can perform the following manipulation.

$$\lim_{|j-i|\to\infty} \left(\mathbf{B}_{i\bullet}\right) = \lim_{|j-i|\to\infty} \left(-\left|\frac{j-i}{\alpha}\right|^\beta\right) = \lim_{|j-i|\to\infty} \left(-\left|\frac{\alpha}{j-i}\right|^{|\beta|}\right) = 0$$

When we plug 0 back into the scoring function we see that it becomes $-\infty$, and consequently the softmax becomes 0.

$$\lim_{|j-i|\to\infty} p_{ij} = \lim_{|j-i|\to\infty} \text{softmax}\left(\mathbf{q}_i\mathbf{K}^\top + \mathbf{M}_{i\bullet} + \mathbf{B}_{i\bullet}\right)$$
$$= \text{softmax}\left(\mathbf{q}_i\mathbf{K}^\top + \mathbf{M}_{i\bullet}\right)$$

Since the result of the softmax is not necessarily zero, $p_{ij}$ also is not necessarily 0, thus GGD-BAM can attend to arbitrarily long context when $\beta < 0$ and $\alpha > 0$. $\qquad\square$

## C  EXPERIMENTAL SETUP

Here we detail the experimental setup necessary to replicate our results.

Our models were based on Llama3 (Grattafiori et al., 2024). We used RMSNorm (Zhang & Sennrich, 2019) and, on feedforward blocks, we used swiglu activation function (Shazeer, 2020). We trained all our models on the Fineweb 10B dataset (Penedo et al., 2024) utilizing the Mistral-7B v0.3 tokenizer (Jiang et al., 2023) for text processing. Training was performed using context lengths of 512, 1024, and 2048 tokens to evaluate performance under different sequence lengths.

To maintain document separation within packed sequences, we employed an attention mask preventing self-attention between distinct documents (Grattafiori et al., 2024). The models were optimized using RAdam with decoupled weight decay set to 0.1 and an initial learning rate of $1 \times 10^{-3}$. We applied a cosine learning rate decay schedule, reducing the learning rate to a minimum of $0.1\times$ its initial value.

We trained LMs up to 1.1 billion learnable parameters. Table 1 details the configuration of each trained model, including embedding, trainable parameters, attention heads, and hidden layer size.

Table 1: Architecture details of the LM we evaluated in our study.

| Attribute | 120M | 432M | 1.1B |
|---|---|---|---|
| Parameters embedding | ~25M | ~50M | ~67M |
| Parameters transformer | ~95M | ~380M | ~1B |
| Parameters BAM ($\theta_\alpha, \theta_\beta$ and $\theta_\mu$) | 576 | 1008 | 1440 |
| Attention heads | 16 | 24 | 32 |
| Layers | 12 | 14 | 15 |
| Hidden size | 768 | 1536 | 2048 |
| ff hidden size | 2×768 | 2×1536 | 4×2048 |
| Learning Rate | $1 \times 10^{-3}$ | $5 \times 10^{-4}$ | $3 \times 10^{-4}$ |

Training was executed for approximately $19,251$ steps with a global batch size of $589,824$ tokens, leveraging up to 6 NVIDIA A6000 GPUs. For the 1.1B parameter models, we perform additional 256 training steps with a context length of 1024. This was necessary because bigger models tends to overfit in the trained context length (see Appendix H.5 for further details). The resulting models from these specific training configurations formed the basis for our subsequent performance evaluation.

In our implementation we define three learnable parameters, $\theta_\mu, \theta_\alpha$ and $\theta_\beta$ for each *attention head*. So $\mathbf{B} = [b_{ij}]_{L \times L}$ where:

$$b_{ij} = -e^{\theta_\alpha}\left(\left|(j-i) - (e^{\theta_\mu} - e^{-\theta_\mu})\right| + \epsilon\right)^{\theta_\beta},$$

for $i = 1, \ldots, L$ and $j = 1, \ldots, L$. So the GGD is parametrized in the following way: $\mu = e^{\theta_\mu} - e^{-\theta_\mu}$, $\beta = \theta_\beta$ and $\alpha = e^{-\frac{\theta_\alpha}{\theta_\beta}}$, that ensures $\alpha \geq 0$. $\epsilon = 10^{-5}$ avoids division by zero when $\beta < 0$.

To evaluate our models in contexts longer than 10,000 tokens we implemented key-value caching (Kim et al., 2023).

# D PERPLEXITY EVALUATION

## D.1 PERPLEXITY ON SHORT CONTEXT

We evaluate generalization for models with distinct PE methods by computing the perplexity for models trained on a context length of 512 in a hold-out validation set of Fineweb with sentences up to 512 tokens and on Wikipedia with sentences up to 512 tokens. Results are shown in Table 2.

Table 2: Perplexity of 120M models in Wikipedia and Fineweb data.

|  | Wikipedia | Fineweb |
|---|---|---|
| Sinusoidal | 18.9310 | 22.3573 |
| ALiBi | 18.5452 | 22.1771 |
| NoPE | 19.8609 | 23.7898 |
| BAM | 18.9281 | 22.2640 |
| RoPE | 18.3599 | 22.4428 |
| Sinusoidal SSMax | 19.1467 | 22.3150 |
| NoPE SSMax | 20.3899 | 23.7581 |
| ALiBi SSMax | 18.4967 | 22.2125 |
| BAM SSMax | 18.6897 | 22.1363 |
| RoPE SSMax | 20.1854 | 24.4507 |

We see that in context length seen during training, BAM outperforms almost all the compared baselines. The only exception is ALiBi, which outperforms BAM by less than 0.1 points in perplexity.

ALiBi outperforming BAM in this case is expected from our probabilistic interpretation of PE. ALiBi is a Laplace distribution only focused on the local context whereas BAM has also the ability to use information in long context. We notice in experiments focused on long context (see Appendices F and H) that this small gap in perplexity to ALiBi translates into huge long-context retrieval gain.

## D.2 PERPLEXITY ON LONG CONTEXT

We evaluate context length generalization for models with distinct PE methods by computing the perplexity for models that were trained on a context length of 512 in a hold-out validation set of sentences longer than 512. This is a similar evaluation procedure as performed by Press et al. (2022).

Results shown in Figure 8 are consistent with those reported in the ALiBi paper (Press et al., 2022), confirming that Sinusoidal, RoPE, and NoPE fail to extrapolate to sequence lengths beyond those seen during training. For these models, perplexity increases sharply as the context grows, indicating that the models are unable to maintain coherent predictions over long sequences. In contrast, only ALiBi, BAM, BAM combined with Scalable Softmax (SSMax), and RoPE local maintain a stable perplexity profile under context extrapolation. These models exhibit sub-linear or nearly flat perplexity growth as the input length increases to 32,000 tokens—despite being trained only on context length of 512 tokens. This confirms our theoretical results where priors introduced by BAM provide robust generalization to unseen context lengths, on par with ALiBi's linear bias-based extrapolation.

We evaluate all the PE context length extrapolation regarding perplexity on Wikipedia dataset. Figure 9 shows the log-scaled perplexity and we see a similar trend to Fineweb 10B. BAM, RoPE and ALiBi are able to maintain low perplexity across all the evaluated context lengths. We see that SSMax has more impact in lowering BAMs perplexity when compared to other PEs.

However, this type of perplexity-based evaluation has limitations (Hu et al., 2024). It does not measure whether the model attends to the full sequence or only relies on the most recent tokens. For instance, the RoPE Local variant implemented here applies local attention restricted to a sliding window and still achieves competitive results with ALiBi. This indicates that models can make accurate next-token predictions without integrating information from earlier parts of the sequence. Hence, we claim that perplexity should not be taken as the sole measure of effective context length extrapolation. Indeed, information retrieval evaluation seems to be more suitable to assess extrapolation of trained lengths.

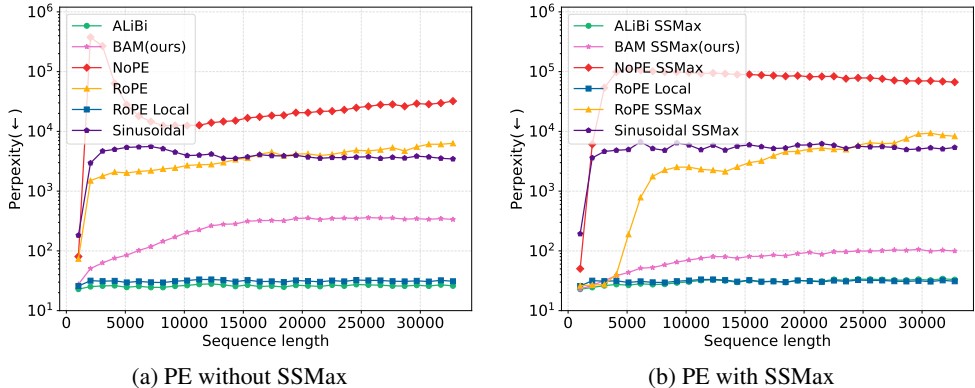

Figure 8: Log-scaled perplexity computed up to $64\times$ the training context length of 512 tokens. BAM, RoPE Local and ALiBi are able to maintain the lowest perplexity on longer contexts.

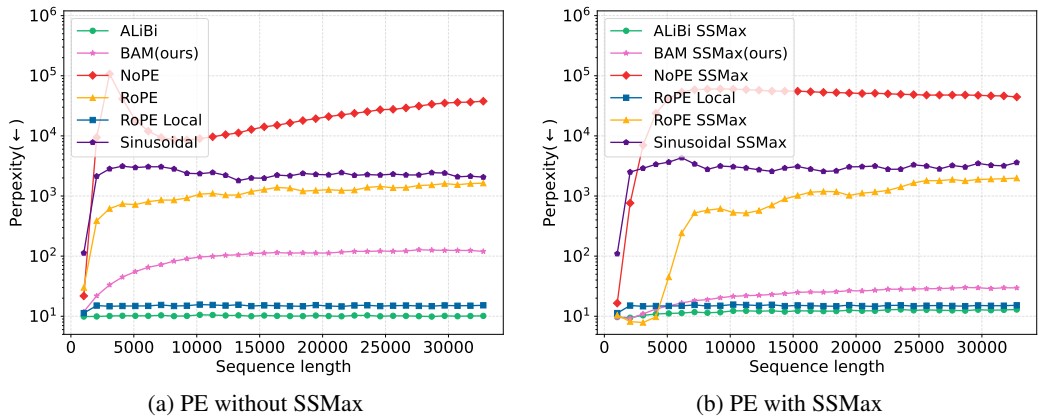

Figure 9: Log-scaled perplexity on Wikipedia dataset. BAM, RoPE Local and ALiBi are able to maintain the lowest perplexity on longer contexts.

## E    DOWNSTREAM EVALUATION

While perplexity provides a general notion regarding model capability in language modeling, it does not necessarily correlate with capability on downstream tasks. Here we evaluate the large scale 1B parameter models performance on downstream tasks from MMLU Hendrycks et al. (2020), ARC-easy, and ARC-challenge benchmarks Clark et al. (2018). Table 3 shows that BAM SSMax is superior to RoPE in all the evaluated benchmarks.

Table 3: GGD-BAM vs RoPE Large-Scale 1B parameter models on MMLU and ARC Benchmarks

|            | MMLU   | ARC-Easy | **ARC-Challenge** |
|------------|--------|----------|-------------------|
| BAM SSMax  | **0.3716** | **0.5770** | **0.4132**        |
| RoPE SSMax | 0.3573 | 0.5715   | 0.4123            |

## F    RULER BENCHMARK

To assess GGD-BAM capability to attend to long-context information, we assessed its performance in the NIAH subset of the Ruler benchmark (Hsieh et al., 2024). We deliberately chose only the NIAH

subset because it is designed to isolate the PE's ability to access information at a specific distance while ensuring the model maintains stable next-token prediction performance.

Performance on other tasks, such as Question Answering and Variable Tracking, is known to correlate more with model size than with context extrapolation, and thus fall out of the scope of this paper.

### F.1 GGD-BAM vs Baselines (120M)

As seen in Table 4, GGD-BAM outperforms all the baselines on the three variations of the NIAH task. Specially in NIAH Single 1, our PE method outperforms all other PE methods by a substantial margin, being the only one able to perform retrieval above $6k$ tokens.

The second best performing PE is RoPE, followed by ALiBi. In the Single 1 version of Ruler NIAH, RoPE maintains $0.64$ accuracy while ALiBi scores a mere $0.16$.

Regardless of the PE, in the Single 3 version of NIAH, where the Passkey appears in the form of a UUID, only BAM achieves an accuracy above $0.8$.

Table 4: Accuracy of 120M models on the NIAH subset of the Ruler Benchmark.

| Task | PE | 1K | 1.5K | 2K | 3K | 4K | 6K | 8K | 10K | 12K |
|---|---|---|---|---|---|---|---|---|---|---|
| Single 1 | NoPE SSMax | 0.04 | 0.00 | 0.00 | 0.00 | 0.00 | 0.00 | 0.00 | 0.00 | 0.00 |
| | RoPE SSMax | 1.00 | 1.00 | 1.00 | 1.00 | 0.64 | 0.00 | 0.00 | 0.00 | 0.00 |
| | ALiBi SSMax | 0.96 | 0.26 | 0.22 | 0.18 | 0.16 | 0.10 | 0.02 | 0.04 | 0.02 |
| | RoPE Local | 0.40 | 0.22 | 0.24 | 0.20 | 0.12 | 0.08 | 0.00 | 0.02 | 0.00 |
| | Sinusoidal SSMax | 0.00 | 0.00 | 0.00 | 0.00 | 0.00 | 0.00 | 0.00 | 0.00 | 0.00 |
| | **BAM SSMax (ours)** | **1.00** | **1.00** | **1.00** | **1.00** | **1.00** | **1.00** | **0.98** | **0.92** | **0.88** |
| Single 2 | NoPE SSMax | 0.70 | 0.00 | 0.00 | 0.00 | 0.00 | 0.00 | 0.00 | 0.00 | 0.00 |
| | RoPE SSMax | 0.98 | 0.96 | 0.94 | 0.84 | **0.52** | 0.00 | 0.00 | 0.00 | 0.00 |
| | ALiBi SSMax | 1.00 | 0.46 | 0.12 | 0.10 | 0.06 | 0.02 | 0.00 | 0.00 | 0.00 |
| | RoPE Local | 0.84 | 0.36 | 0.10 | 0.14 | 0.10 | 0.00 | 0.00 | 0.00 | 0.00 |
| | Sinusoidal SSMax | 0.70 | 0.00 | 0.00 | 0.00 | 0.00 | 0.00 | 0.00 | 0.00 | 0.00 |
| | **BAM SSMax (ours)** | **1.00** | **1.00** | **1.00** | **0.88** | 0.24 | **0.06** | **0.02** | 0.00 | 0.00 |
| Single 3 | NoPE SSMax | 0.04 | 0.00 | 0.00 | 0.00 | 0.00 | 0.00 | 0.00 | 0.00 | 0.00 |
| | RoPE SSMax | 0.28 | 0.30 | 0.10 | 0.06 | 0.00 | 0.00 | 0.00 | 0.00 | 0.00 |
| | ALiBi SSMax | 0.24 | 0.04 | 0.00 | 0.00 | 0.00 | 0.00 | 0.00 | 0.00 | 0.00 |
| | RoPE Local | 0.10 | 0.00 | 0.00 | 0.00 | 0.00 | 0.00 | 0.00 | 0.00 | 0.00 |
| | Sinusoidal SSMax | 0.08 | 0.00 | 0.00 | 0.00 | 0.00 | 0.00 | 0.00 | 0.00 | 0.00 |
| | **BAM SSMax (ours)** | **0.84** | **0.68** | **0.42** | **0.08** | 0.00 | 0.00 | 0.00 | 0.00 | 0.00 |

### F.2 Large Scale GGD-BAM vs RoPE

Here we perform a large-scale experiment directly comparing RoPE and BAM with 1B parameters in the Ruler benchmark. Results are show in Table 5. BAM achieves superior performance in comparison to RoPE in every evaluated task in the Ruler benchmark, with a highlight of achieving almost perfect accuracy across tasks on the Single 1 subset.

Other tasks such as Multikey 2 and Multikey 3 are harder for both models. This shows that only our pre-training may not be enough for models to perform such tasks. However, since our goal here is to assess how distinct PE behave on exactly the same training regime, we see that GGD-BAM clearly achieves better longer context in Single 1, 2 and 3, MultiKey 1, MultiQuery and MultiValue.

## G LongBenchv2

In Table 6, we present the results of 1B parameter models on the complete LongBenchV2 benchmark Bai et al. (2025) limited in 131k tokens. We chose to report just the large scale 1B parameter models because smaller models perform close to random guessing in these tasks. BAM outperforms

Table 5: GGD-BAM vs RoPE Ruler Benchmark Large-Scale 1B parameter models.

| Task | PE | 1024 | 1536 | 2048 | 3072 | 4096 | 6144 | 8192 | 12288 | 16384 | 24576 | 32768 |
|------|-----|------|------|------|------|------|------|------|-------|-------|-------|-------|
| Single 1 | BAM SSMax | **1.00** | **1.00** | **1.00** | **1.00** | **1.00** | **1.00** | **1.00** | **1.00** | **1.00** | **1.00** | **1.00** |
| | RoPE SSMax | **1.00** | 0.88 | 0.68 | 0.40 | 0.30 | 0.10 | 0.02 | 0.00 | 0.00 | 0.00 | 0.00 |
| Single 2 | BAM SSMax | **1.00** | **1.00** | **1.00** | 0.98 | **1.00** | 0.88 | 0.82 | 0.46 | 0.18 | 0.06 | 0.02 |
| | RoPE SSMax | **1.00** | 0.82 | 0.62 | 0.32 | 0.18 | 0.04 | 0.00 | 0.00 | 0.00 | 0.00 | 0.00 |
| Single 3 | BAM SSMax | 0.88 | 0.92 | 0.88 | 0.86 | 0.80 | 0.62 | 0.30 | 0.10 | 0.02 | 0.00 | 0.00 |
| | RoPE SSMax | 0.76 | 0.30 | 0.16 | 0.04 | 0.02 | 0.00 | 0.00 | 0.00 | 0.00 | 0.00 | 0.00 |
| MultiKey 1 | BAM SSMax | 0.84 | 0.86 | 0.94 | 0.92 | 0.86 | 0.76 | 0.66 | 0.56 | 0.24 | 0.10 | 0.06 |
| | RoPE SSMax | 0.80 | 0.86 | 0.68 | 0.38 | 0.32 | 0.08 | 0.06 | 0.00 | 0.00 | 0.00 | 0.00 |
| MultiKey 2 | BAM SSMax | 0.22 | **0.16** | **0.12** | **0.04** | **0.04** | 0.00 | **0.02** | **0.02** | 0.00 | 0.00 | 0.00 |
| | RoPE SSMax | **0.26** | 0.14 | 0.04 | 0.00 | 0.00 | 0.00 | 0.00 | 0.00 | 0.00 | 0.00 | 0.00 |
| MultiKey 3 | BAM SSMax | 0.12 | **0.04** | 0.00 | 0.00 | 0.00 | 0.00 | 0.00 | 0.00 | 0.00 | 0.00 | 0.00 |
| | RoPE SSMax | **0.18** | 0.00 | 0.00 | 0.00 | 0.00 | 0.00 | 0.00 | 0.00 | 0.00 | 0.00 | 0.00 |
| MultiQuery | BAM SSMax | 0.88 | 0.88 | 0.85 | 0.82 | 0.71 | 0.68 | 0.62 | 0.44 | 0.23 | 0.10 | 0.02 |
| | RoPE SSMax | 0.85 | 0.76 | 0.42 | 0.17 | 0.11 | 0.06 | 0.03 | 0.03 | 0.01 | 0.01 | 0.01 |
| MultiValue | BAM SSMax | 0.96 | 0.94 | 0.92 | **0.92** | **0.82** | 0.84 | 0.76 | 0.70 | 0.38 | 0.14 | 0.00 |
| | RoPE SSMax | 0.96 | **1.00** | **1.00** | 0.78 | 0.70 | 0.42 | 0.32 | 0.10 | 0.04 | 0.08 | 0.00 |

RoPE in all evaluated tasks, with an overall score 5 points above RoPE. We opted not to show the Long Structured Data Understanding task because it has only four instances under 131k tokens.

Table 6: LongBenchv2 Benchmark: GGD-BAM vs RoPE, 1B parameter models.

| | BAM SSMax | RoPE SSMax |
|---|-----------|------------|
| Code Repository Understanding | **41.7** | 25.0 |
| Long In-context Learning | **36.4** | 30.3 |
| Long-dialogue History Understanding | 35.0 | 35.0 |
| Multi-Document QA | **26.5** | 25.3 |
| Single-Document QA | **26.5** | 18.8 |
| Overall | **28.6** | 24.2 |

# H   ABLATION STUDY

## H.1   INITIALIZATION

In this section we study two different initialization strategies for the shape $\theta_\beta$, scale $\theta_\alpha$ and location $\theta_\mu$ parameters of GGD-BAM. The first initialization is a Laplacian that replicates ALiBi, setting $\theta_\beta = 1$, different $\theta_\alpha$ for each layer and $\theta_\mu = 0$. The second initialization start from Uniform distribution prior $\theta_\beta = 0$, which is a middle ground between ALiBis Laplacian and $\theta_\beta < 0$, with $\theta_\alpha = 0$.

Figure 10 shows us that ALiBi initialization provides least extrapolation lengths. Our best results were achieved by using the initialization scheme of $\theta_\beta = 0$ and $\theta_\alpha = 0$, this initialization is equivalent to assigning a Uniform prior to all tokens in the context.

## H.2   LEARNABLE PARAMETERS

The GGD prior in BAM is parameterized by a shape parameter $\theta_\beta$, a scale parameter $\theta_\alpha$, and a location parameter $\theta_\mu$. In this section, we evaluate how training different subsets of these parameters affects the performance of GGD-BAM. Each configuration introduces a different number of additional trainable parameters: training only $\theta_\beta$ adds 192 parameters; training both $\theta_\beta$ and $\theta_\alpha$ adds 384; and training all three parameters ($\theta_\beta$, $\theta_\alpha$, and $\theta_\mu$) adds 576 parameters to the model.

In Figure 11 we can see that allowing all parameters $\theta_\beta, \theta_\alpha$ and $\theta_\mu$ to be learned during training lowers model capacity to extrapolate. The best result was achieved when both $\theta_\beta$ and $\theta_\alpha$ are learn during training, showing that both parameters are important for context length extrapolation.

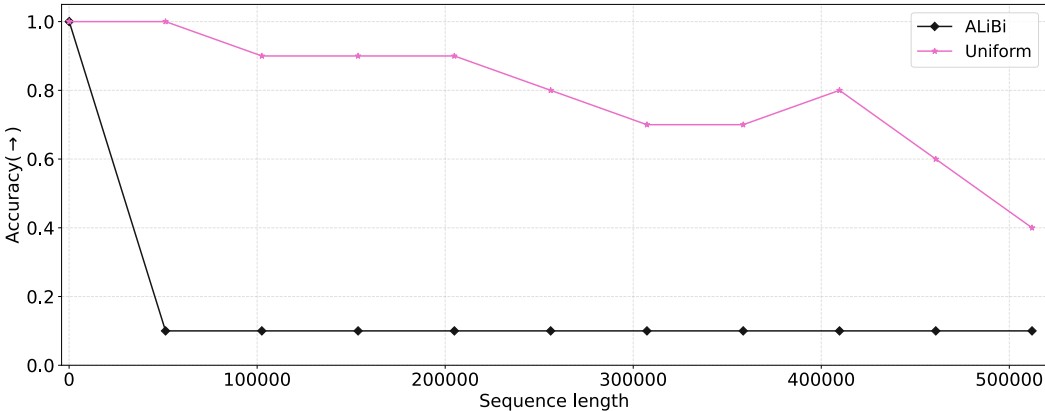

Figure 10: Comparison of passkey retrieval accuracy of models trained on context length 512 training just $\theta_\beta$ and $\theta_\alpha$ with three distinct parameter initialization schemes.

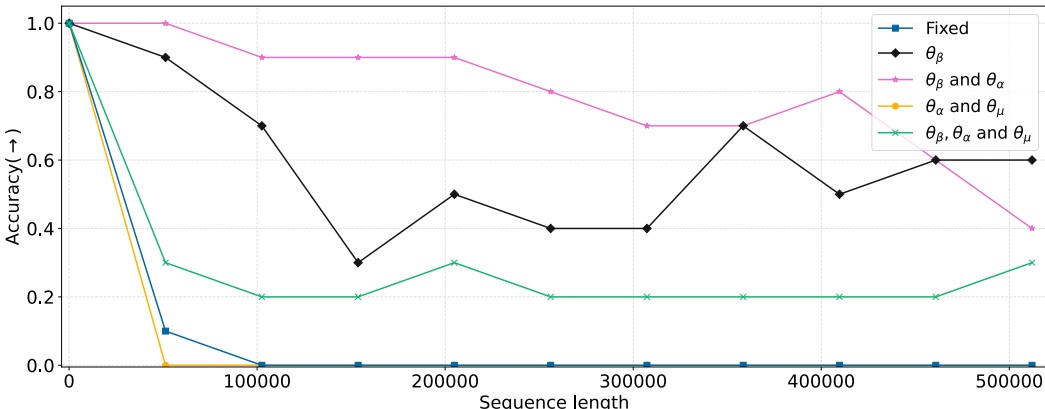

Figure 11: Passkey retrieval accuracy of models training just: $\theta_\beta$; $\theta_\beta$ and $\theta_\alpha$; and $\theta_\beta$, $\theta_\alpha$ and $\theta_\mu$. All variations were trained on context length 512. Training $\theta_\beta$ and $\theta_\alpha$ while fixing $\theta_\beta = 0$ yields more extrapolation length.

If we compare results on Figure 11 to other PE in Figure 3 we see that even our worst combination of training all parameters is superior to all other PE in long context passkey retrieval.

### H.3 TRAINING CONTEXT LENGTH

Here we repeat the experiments for models trained on context length of 1,024 and 2,048, double and quadruple the original context length of 512. We show detailed results for training context length of 1,024 both on perplexity and passkey retrieval when compared to other PE methods. And we also compare passkey retrieval accuracy between BAM SSMax on those three distinct context lengths.

Essentially, the trend of ALiBi, RoPE Local and BAM SSMax being the only PE methods that are able to maintain low perplexity on longer context is maintained, this is possible to identify in Figure 12. As expected, NoPE and Sinusoidal PE are the first models to exponentially increase in perplexity.

RoPE SSMax improved it extrapolation performance and was able to maintain perplexity on par with ALiBi until around 7,500 tokens. It is worth noting that to achieve such results with RoPE, we expanded the $\mathbf{R}_\theta$ manipulation post-training that was performed by Nakanishi (2025) from $50\times$ to $100\times$. Without this RoPE SSMax would perform on par with standard RoPE.

Regarding passkey retrieval accuracy, again all PE methods except BAM SSMax struggles to access long context information and maintain high accuracy, Figure 13 shows this trend. Although the Figure 13 shows context lengths up to 32,000, we evaluated BAM SSMax until 512,000 context

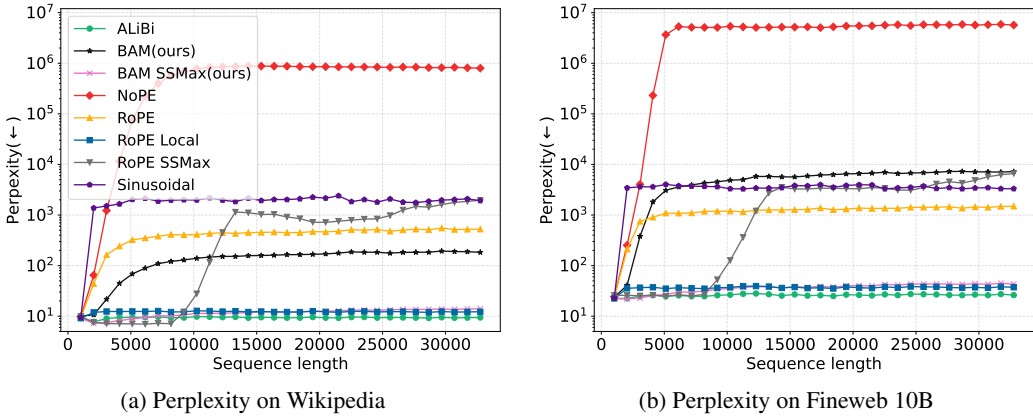

(a) Perplexity on Wikipedia    (b) Perplexity on Fineweb 10B

Figure 12: Log-scaled perplexity computed up to $32\times$ the training context length of 1024 tokens. BAM SSMax, RoPE Local and ALiBi are able to maintain the lowest perplexity on longer contexts.

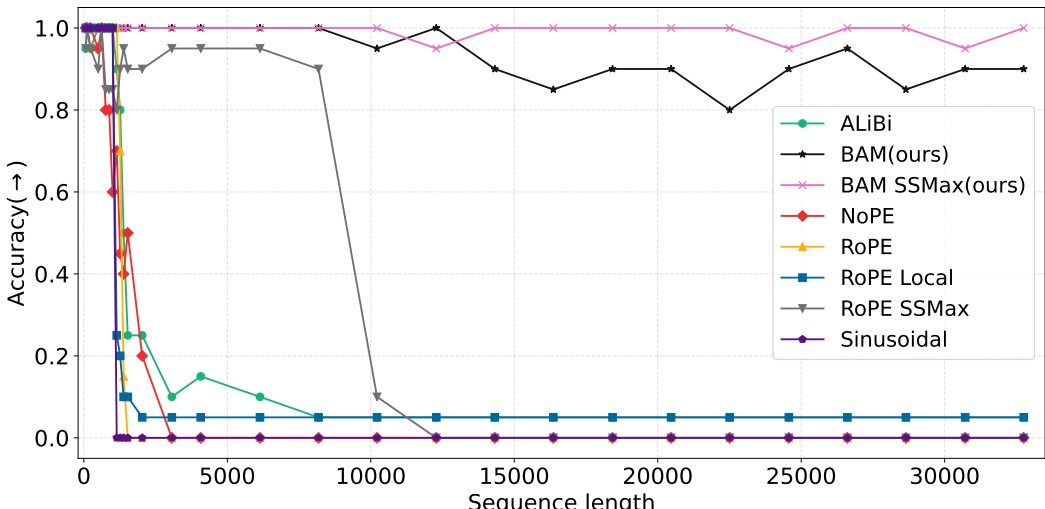

Figure 13: Passkey retrieval accuracy of models with distinct PE in the Passkey Retrieval task. BAM and BAM SSMax outperform all PE methods. Specifically BAM SSMax is capable of maintaining perfect accuracy for a context $32\times$ the training context length.

length and it maintained accuracy above 80% until 300,000 and did not drop to zero throughout the evaluation. Beyond 512,000 context length, we did not have enough vram to perform evaluation.

In Figure 14 we can see how training in longer context lengths affects GGD-BAM. Generally, training for longer contexts appears to make the model more robust to long context generalization as the accuracy tend to drop slower. The model trained with context length of $2,048$ tokens generalizes all the way to 512k tokens while maintaining accuracy above $90\%$. The model trained with context length of $512$ achieves $40\%$ accuracy on 512k, showing a correlation between trained context length and retrieval accuracy at 512k tokens. Nevertheless, our model trained on context length of 512 is competitive with all the others until $300,000$ where others achieve higher accuracy.

## H.4   SCALABLE SOFTMAX

Here we test all PE methods without SSMax. Figure 15 shows the same trend of models with SSMax, BAM outperforms every other PE method. It is worth noting that SSMax improves context length

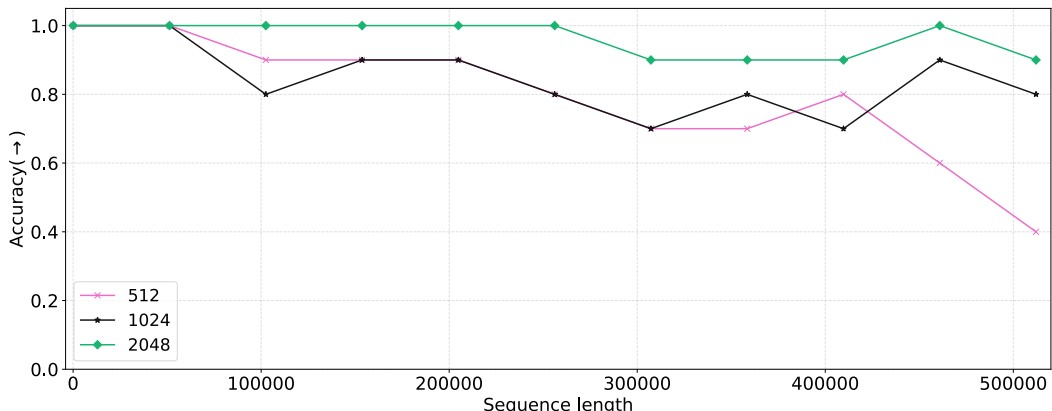

Figure 14: Comparison of passkey retrieval accuracy of GGD-BAM models trained on context length 512, 1,024 and 2,048.

generalization in almost all the assessed PE methods. This shows that *fading attention* is indeed one of the problems in long context extrapolation, however BAM is superior to other PE on both cases.

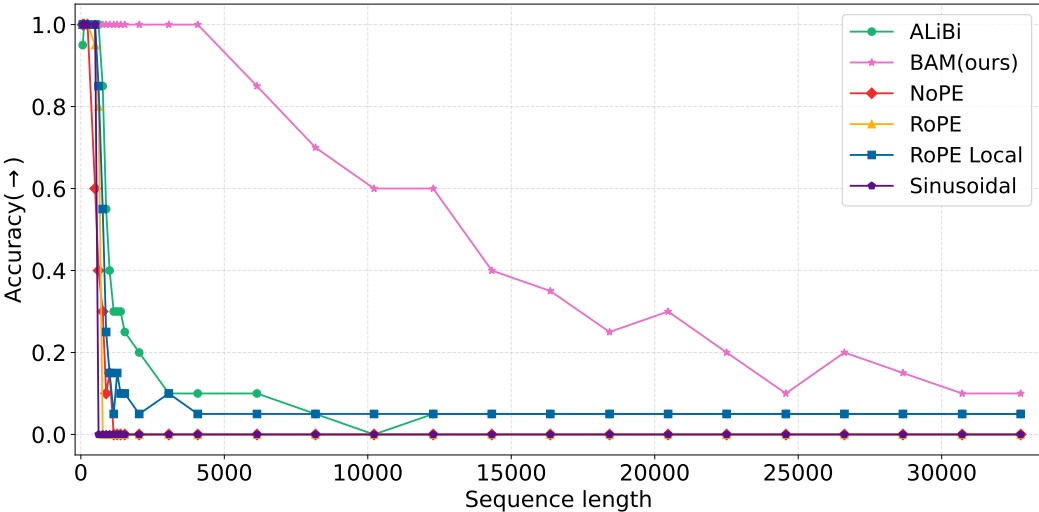

Figure 15: Passkey retrieval accuracy with distinct PE. BAM outperform all PE methods.

By comparing the results from BAM and BAM SSMax, we note that SSMax plays a role in maintaining the PE ability to extrapolate to longer lengths. This trend is also maintained when comparing RoPE and its SSMax version. The fact that Scalable Softmax improves context length generalization both for BAM and for RoPE shows that good PE is necessary but not sufficient for context length extrapolation. The softmax function tends to zero for longer context windows, which is a problem both for language modeling and for retrieval. To counterbalance this effect, scalable softmax applies a rescaling factor to the logits (Nakanishi, 2025), fixing that limitation.

Scalable Softmax introduces a rescaling factor $s \times \ln(n)$, where $n$ is the size of the input vector and $s$ is a learnable parameter. Note that $s$ has a similar effect to the normalizing scaler $Z$ obtained when framing PE as a Bayesian mechanism. The only effective difference is that $Z$ should be a function of both query and keys whereas $s$ is a learnable parameter.

To understand the effect of the normalizing scalar $s$ in our models across distinct scales, we show in Figure 16 how this learnable parameter is distributed after training. We see that although the distribution appears to have a heavier tail in smaller models, the shape of the distribution is similar across model scales.

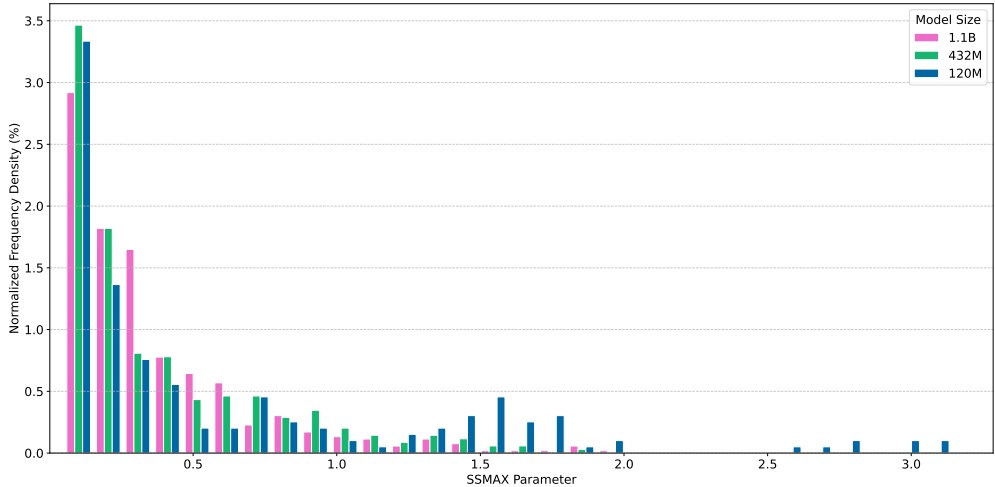

Figure 16: Scaling factor $s$ in scalable softmax in three distinct model scales.

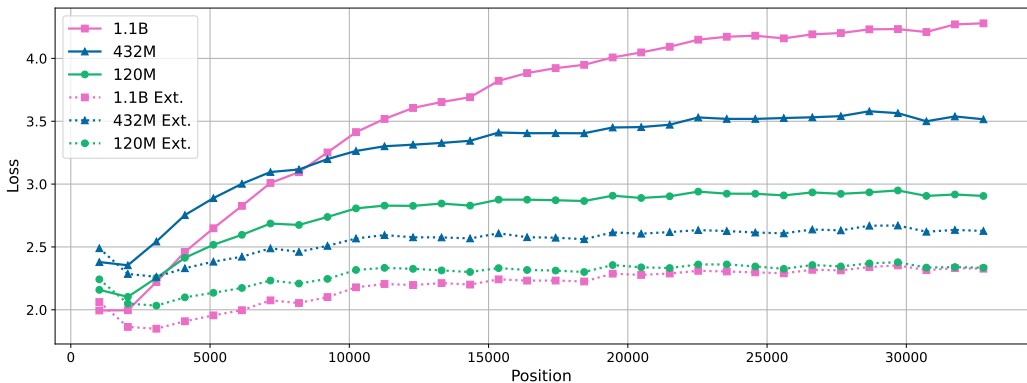

Figure 17: Loss of GGD-BAM before and after lightweight fine-tuning for context extension.

## H.5 CONTEXT EXTENSION

We observed during our first experiments that smaller models generalized better to longer context. We found out that bigger models are prone to overfitting the trained context length.

We devised a lightweight fine-tuning with context length $1024$ for $256$ steps (in comparison to $512$ during the beginning of the training) to make bigger models generalize once again. In Figure 17, we show that after lightweight fine-tuning the loss of our models become more stable for context beyond the training length of $512$ tokens. This shows that, even if bigger models can overfit the training context length, a lightweight fine-tuning procedure can make them generalize to extended contexts.

After context extension, we performed the PassKey Retrieval analysis and noticed that bigger models benefit more from the context extension than smaller ones. The results obtained by this lightweight fine-tuning procedure are shown in Table 7.

In Table 7 we see that the 120M model generalizes up to $512\times$ the trained context length with accuracy above $0.8$. However, bigger models struggle in much shorter sequences. When analyzing each model to their context-extended counterparts, we see that all model-scales benefit from this procedure. The 1.1B parameter model, however, has most improvement of context extension.

Table 7: Context extension effect across different model sizes of BAM SSMax on PassKey Retrieval.

| Model | <1K | 51K | 102K | 153K | 204K | 256K | 307K | 358K | 409K | 460K | 512K |
|---|---|---|---|---|---|---|---|---|---|---|---|
| 120M | 1.0 | 1.0 | 0.9 | 0.9 | 0.9 | 0.8 | 0.7 | 0.7 | 0.8 | 0.6 | 0.4 |
| 120M Ext. | 1.0 | 1.0 | 1.0 | 1.0 | 1.0 | 1.0 | 1.0 | 1.0 | 1.0 | 1.0 | 1.0 |
| 431M | 1.0 | 0.5 | 0.1 | 0.3 | 0.1 | 0.1 | 0.1 | 0.2 | 0.1 | 0.0 | 0.2 |
| 431M Ext. | 1.0 | 1.0 | 1.0 | 1.0 | 1.0 | 1.0 | 1.0 | 1.0 | 1.0 | 1.0 | 1.0 |
| 1.1B | 0.9 | 0.0 | 0.0 | 0.0 | 0.0 | 0.1 | 0.0 | 0.0 | 0.1 | 0.0 | 0.0 |
| 1.1B Ext. | 1.0 | 1.0 | 1.0 | 1.0 | 1.0 | 0.9 | 1.0 | 0.8 | 0.9 | 0.7 | 0.7 |

## H.6 MODEL SIZE

Here we compare how BAM SSMax compares to RoPE SSMax on a large scale model size (1.1B) on the Passkey Retrieval setting. Models were trained with context length 512 and prompted to perform Passkey Retrieval with up to $512,000$ tokens. Figure 18 shows that BAM SSMax also dominates RoPE SSMax across all evaluated lengths, performing accurate Passkey retrieval in all contexts that we were able to assess using our available compute.

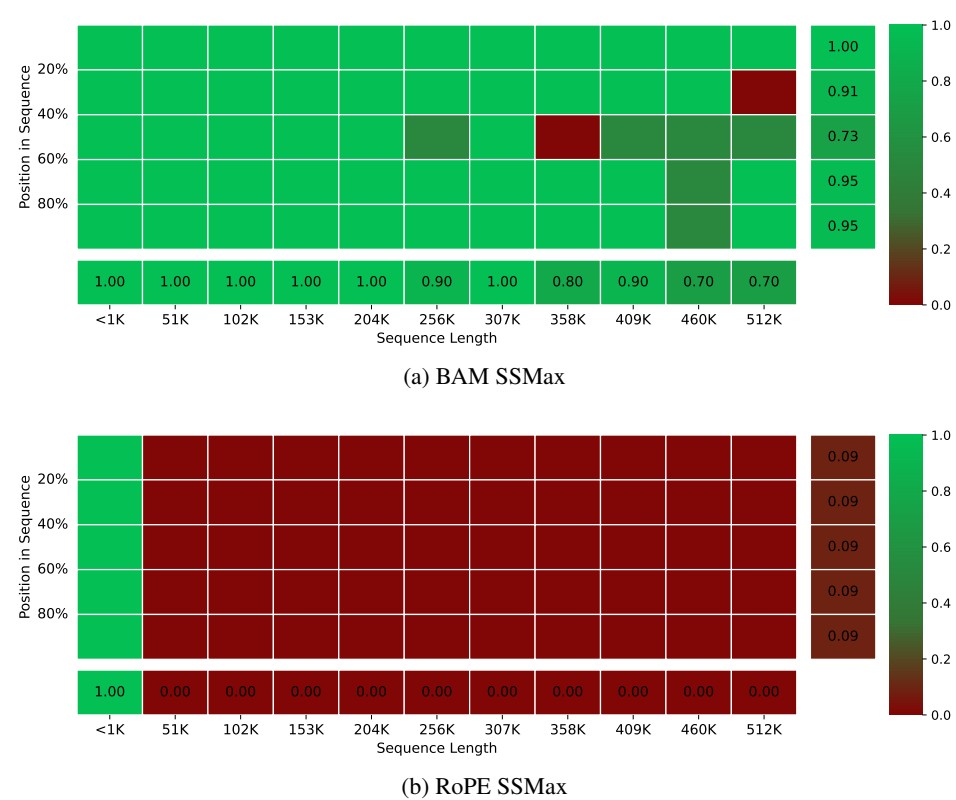

(a) BAM SSMax

(b) RoPE SSMax

Figure 18: Passkey Retrieval accuracy on 1.1B models. In the bottom row and on the last column, we see average accuracy across length and position, respectively.

This ablation corroborates that BAM is capable of using information across longer contexts than RoPE, and that such a conclusion generalizes across bigger models (and not just in 120M settings).

## H.7 PE IMPACT ON INFERENCE PERFORMANCE

Here we access how distinct PE strategies impact model throughput during inference. To perform this experiment, we initialize BAM and all the baselines trainable weights of four distinct model sizes and run 100 samples with batch size 1 and sequence length 512.

In Table 8, we see that for smaller models the cost of performing Scalable Softmax dominates the results. This is imperceptible in bigger models, since the only impact of Scalable Softmax appears to be in the standard deviation. BAM does not affect model inference time in comparison to other PEs. When we account for the standard deviation, every model has equivalent inference time.

Table 8: Inference time (ms) and vram (GB) during the backward pass for distinct model sizes.

| | 120M | | 430M | | 1.1B | |
| --- | --- | --- | --- | --- | --- | --- |
| | Time | VRAM | Time | VRAM | Time | VRAM |
| Sinusoidal | $32.84 \pm 0.40$ | 1.734 | $50.98 \pm 0.37$ | 4.933 | $117.26 \pm 0.42$ | 12.329 |
| Sinusoidal SSMax | $32.82 \pm 1.13$ | 1.736 | $53.39 \pm 0.28$ | 4.933 | $120.91 \pm 0.55$ | 12.324 |
| RoPE | $35.77 \pm 1.98$ | 1.754 | $53.42 \pm 0.38$ | 4.936 | $120.50 \pm 0.36$ | 12.330 |
| RoPE SSMax | $36.04 \pm 1.39$ | 1.742 | $55.65 \pm 0.17$ | 4.934 | $123.34 \pm 0.42$ | 12.325 |
| ALiBi | $31.43 \pm 0.73$ | 1.738 | $52.24 \pm 0.27$ | 4.933 | $119.34 \pm 0.39$ | 12.329 |
| ALiBi SSMax | $32.21 \pm 0.09$ | 1.736 | $54.22 \pm 0.21$ | 4.934 | $122.09 \pm 0.33$ | 12.325 |
| BAM | $33.13 \pm 0.53$ | 1.739 | $52.85 \pm 0.24$ | 4.936 | $120.31 \pm 0.38$ | 12.329 |
| BAM SSMax | $33.01 \pm 0.81$ | 1.737 | $59.50 \pm 0.18$ | 4.936 | $133.14 \pm 0.41$ | 12.319 |
| NoPE | $38.15 \pm 1.87$ | 1.738 | $56.10 \pm 0.42$ | 4.933 | $121.34 \pm 0.80$ | 12.329 |
| NoPE SSMax | $38.49 \pm 1.30$ | 1.736 | $58.36 \pm 0.70$ | 4.933 | $124.95 \pm 0.60$ | 12.324 |

## I LIMITATIONS

Despite the theoretical and empirical strengths of BAM and its instantiation with GGD, our study is subject to several limitations, which we acknowledge and discuss below.

**Model Scale and Generalization to Larger LMs.** Our experiments were conducted on Transformer models with up to 1.1 billion parameters due to limited compute availability. While our results show improvements in context length extrapolation at this scale, it remains an open question whether these gains persist or even amplify in very large language models. However, we note that this evaluation regime is consistent with prior work in the PE literature, including those introducing ALiBi (Press et al., 2022) and NoPE (Kazemnejad et al., 2023), which also validated their approaches using similar-scale models.

**Dataset Scope and Representativeness.** Our empirical evaluation of perplexity is currently restricted to two datasets: FineWeb 10B (Penedo et al., 2024), and Wikipedia (Foundation, 2023). While these datasets provide coverage of both large-scale pretraining and structured text, this coverage is not exhaustive. Broader evaluations across additional domains—such as code, long-form scientific documents—would be valuable for assessing the robustness and generality of BAM-based priors.

**Coverage of Positional Encoding Methods.** Due to computational constraints and the complexity of reimplementing certain positional encoding strategies, our experiments do not encompass all methods proposed in the literature. We did not evaluate the T5 relative position bias approach (Raffel et al., 2020), which requires bucketing mechanisms and distinct architectural modifications. Nonetheless, we believe that the set of baselines considered—covering absolute positional encodings (Sinusoidal), rotary encodings (RoPE), relative linear biases (ALiBi), and content-only baselines (NoPE)—provides a representative and diverse comparison to assess the context extrapolation capabilities of BAM.

**Generalization to Instruction/Preference Tuned LMs.** We did not evaluate BAM in the context of instruction-tuned or preference-tuned models. LMs often undergo additional fine-tuning stage, such as supervised instruction following, reinforcement learning from human feedback, or direct preference optimization, which can significantly alter attention dynamics and generalization behavior. It remains an open question whether the context extrapolation benefits introduced by BAM are preserved, attenuated, or potentially enhanced in such settings. Assessing the how models with BAM as PE perform after instruction-tuned architectures is an important direction for future work.

## J   BROADER IMPACTS

Improving context length extrapolation in Transformers has the potential to reduce the computational and environmental costs associated with pretraining large language models. Because attention scales quadratically with sequence length, training with long contexts is prohibitively expensive. GGD-BAM enables models to generalize to longer sequences without requiring direct exposure during training, potentially lowering the need for long-context pre-training.

This efficiency gain could contribute to more sustainable and accessible language model development, particularly for institutions with limited compute resources. Furthermore, better long-range generalization supports important applications such as legal and medical document processing, educational content understanding, and scientific analysis. However, these capabilities must be accompanied by careful evaluation to ensure reliability and safety in high-stakes domains.

## K   ALTERNATIVE INTERPRETATIONS OF $Z$

### K.1   STATISTICAL-PHYSICS INTERPRETATION OF $Z$

Let $f_j = f_{\text{cont}}(\mathbf{q}_i, \mathbf{k}_j), \quad g_j = g_{\text{pos}}(i, j).$

We introduce three normalization constants:

$$Z_{\text{cont}} \;=\; \sum_j e^{f_j}, \quad Z_{\text{pos}} \;=\; \sum_j e^{g_j}, \quad Z_{\text{joint}} \;=\; \sum_j e^{f_j + g_j}.$$

The usual *Gibbs–Boltzmann partition function* is

$$Z_{\text{joint}} = \sum_j \exp(f_j + g_j),$$

and the corresponding free energy is $F_{\text{joint}} = -\ln Z_{\text{joint}}.$

The *factorization normalizer* $Z$ that restores $\sum_j \text{softmax}(f_j + g_j) = 1$ after factorizing $\exp(f_j + g_j) = \exp(f_j)\exp(g_j)$ is

$$Z = \frac{Z_{\text{cont}} Z_{\text{pos}}}{Z_{\text{joint}}} = \sum_j \big[\text{softmax}(f_j) \times \text{softmax}(g_j)\big].$$

Its log, $\ln Z = \ln Z_{\text{cont}} + \ln Z_{\text{pos}} - \ln Z_{\text{joint}}$, is precisely the *interaction free-energy* between the *content* and *position* potentials. Finally, introducing an inverse temperature $\gamma$ yields

$$p_{ij} \;\propto\; \exp\big(\gamma\,(f_j + g_j)\big),$$

so $\gamma$ could control how sharply the attention distribution peaks.

### K.2   GEOMETRIC INTERPRETATION OF $Z$

Recall

$$p_{\text{cont}}(j) \;=\; \text{softmax}\big(f_{\text{cont}}(q_i, k_j)\big), \qquad p(g_{\text{pos}}(i, j)) \;=\; \text{softmax}\big(g_{\text{pos}}(i, j)\big),$$

and let

$$p_{\text{cont}} = \big(p(f_{\text{cont}}(\mathbf{q}_i, \mathbf{k}_j))\big)_{j=1}^{n}, \quad p_{\text{pos}} = \big(p(g_{\text{pos}}(i, j))\big)_{j=1}^{n}.$$

Both vectors lie in the probability simplex $\Delta^{n-1} = \{x \in \mathbb{R}^n : x_j \geq 0,\ \sum_j x_j = 1\}$. Then

$$Z = \sum_{j=1}^{n} p(f_{\text{cont}}(\mathbf{q}_i, \mathbf{k}_j)), p(g_{\text{pos}}(i, j)) = \langle p_{\text{cont}}, p_{\text{pos}} \rangle,$$

where $\langle p_{\text{cont}}, p_{\text{pos}} \rangle = \sum_j p(f_{\text{cont}}(\mathbf{q}_i, \mathbf{k}_j)), p(g_{\text{pos}}(i, j)).$

- **Dot-product as overlap.** $\langle p_{\text{cont}}, p_{\text{pos}} \rangle \in [0, 1]$ measures how much the two distributions "agree"—it is maximal when they coincide and minimal when they are disjoint.

- **Norms of probability vectors.** Since $\|p\|_2 \le \|p\|_1 = 1$ for any $p \in \Delta^{n-1}$, the raw dot-product is not a true cosine similarity unless one divides by $\|p_{\text{cont}}\|_2 \|p_{\text{pos}}\|_2$. We omit that division because we need $\sum_j p(f_{\text{cont}}(\mathbf{q}_i, \mathbf{k}_j)), p(g_{\text{pos}}(i, j))$ exactly to quantify the normalization gap of the product of two softmaxes.

- **Re-normalization identity.** The product distribution $p_{\text{cont}} \odot p_{\text{pos}}$ sums to $\langle p_{\text{cont}}, p_{\text{pos}} \rangle \ne 1$. Inverting that sum,

$$Z = \sum_j p(f_{\text{cont}}(\mathbf{q}_i, \mathbf{k}_j)), p(g_{\text{pos}}(i, j)),$$

precisely restores $\sum_j \big[ p(f_{\text{cont}}(\mathbf{q}_i, \mathbf{k}_j)), p(g_{\text{pos}}(i, j)) \big] Z = 1$.

- **Cosine-similarity caveat.** If one instead defined $\cos \theta = \frac{\langle p_{\text{cont}}, p_{\text{pos}} \rangle}{\|p_{\text{cont}}\|_2 \|p_{\text{pos}}\|_2}$, that extra normalization would destroy the simple re-normalization identity needed for attention.

Thus $Z = 1/\langle p_{\text{cont}}, p_{\text{pos}} \rangle$ has a clear geometric meaning: it compensates for the overlap (or misalignment) between the content-based and position-based probability vectors, ensuring their elementwise product yields a valid distribution.

### K.3 INFORMATION-THEORETIC INTERPRETATION OF $Z$

Let

$$p_{\text{joint}}(j) = p_{ij} = \text{softmax}\big(f_{\text{cont}}(q_i, k_j) + g_{\text{pos}}(i, j)\big),$$

and recall the marginals

$$p_{\text{cont}}(j) = \text{softmax}\big(f_{\text{cont}}(q_i, k_j)\big), \quad p_{\text{pos}}(j) = \text{softmax}\big(g_{\text{pos}}(i, j)\big).$$

The mutual information between *content* and *position* under the joint distribution is

$$I(\text{cont}; \text{pos}) = \sum_j p_{\text{joint}}(j) \ln \frac{p_{\text{joint}}(j)}{p(f_{\text{cont}}(\mathbf{q}_i, \mathbf{k}_j)), p_{\text{pos}}(j)}.$$

Since

$$p_{\text{joint}}(j) = \frac{e^{f_j + g_j}}{Z_{\text{joint}}} \quad \text{and} \quad p(f_{\text{cont}}(\mathbf{q}_i, \mathbf{k}_j)), p_{\text{pos}}(j) = \frac{e^{f_j}}{Z_{\text{cont}}} \frac{e^{g_j}}{Z_{\text{pos}}},$$

one shows directly that

$$I(\text{cont}; \text{pos}) = \ln \frac{Z_{\text{cont}} Z_{\text{pos}}}{Z_{\text{joint}}} = \ln Z.$$

Hence

$$Z = \exp\big(I(\text{cont}; \text{pos})\big),$$

which implies:

- $Z = 1$ if and only if content and position are statistically independent ($I = 0$).
- $Z$ grows exponentially with the amount of shared information between content and position.

Thus $Z$ can be seen as the "information-coupling" multiplier that re–normalizes the product of the two marginals into the true joint attention distribution.

