# OpenReview forum: "Bayesian Attention Mechanism: A Probabilistic Framework for Positional Encoding and Context Length Extrapolation"
_ICLR.cc/2026/Conference — ICLR 2026 Poster_

### Official Review · Reviewer_b3U5 · 2025-10-27

**Soundness:** 2
**Presentation:** 3
**Contribution:** 2
**Rating:** 4
**Confidence:** 3

**Summary:**

This paper proposes the Bayesian Attention Mechanism (BAM), framing positional encoding as a Bayesian positional prior that unifies methods like NoPE and ALiBi and introduces GGD-BAM, achieving strong long-context retrieval up to ≈256K tokens with <1000 extra parameters and improved stability when combined with SSMax on models from 120M–1.1B trained on FineWeb10B/Wikipedia.

**Strengths:**

- Formalizes positional encoding as a Bayesian prior, providing a useful theoretical advance.
- Demonstrates robust long-context retrieval (≈256K tokens) with minimal parameter overhead.
- Presents ablations and visualizations that enhance interpretability.
- Supplies clear experimental documentation and reproducibility details in the appendices.

**Weaknesses:**

1. Experiments stop at 1.1B parameters, leaving scaling behavior at 7B/13B+ unknown.
2. Evaluation is narrow (FineWeb, Wikipedia, passkey retrieval, RULER–NIAH) and omits broader benchmarks and domain corpora.
3. Training overhead analysis lacks detailed wall-clock and memory cost measurements for ultra-long fine-tuning.

**Questions:**

1. Is there an empirical correlation between the normalization $Z$ and long-context extrapolation, and can $Z$ be optimized or learned to extend coverage?
2. Does the optimal $β$ shift with larger training contexts (e.g., 2048), and can you provide experiments across different training-context regimes?
3. Why do larger models (1.1B) tend to overfit the training context more, and can you analyze the representation or optimization dynamics behind this?
4. Could a per-layer or learned $β$ schedule better balance local vs. long-range dependencies and reduce local-context suppression when $β<0$?

Overall, while the idea is conceptually interesting and well-presented, the current empirical evidence is insufficient to justify acceptance at this venue.

---

> ### Author Response · Authors · 2025-11-21
>
> # Weakness 1 - Model Scale
> The reviewer is right in his observation that our experiments are restricted to models ranging from $120$M to $1.1$B parameters.
> Our computational budget is small (as pointed in the experimental setup section of our paper). We have access to 4 NVidia A6000 GPUS with 48GB Vram each.
> In our setup, the training of a $1.1$B parameter model takes almost two entire weeks of exclusive allocation in our servers. Training $7$B is a task that would demand almost $420$ days of our compute (taking into account we would need $7\times$ more parameters and $7\times$ more data, a $49\times$ increase in time is a conservative estimate).
>
> We kindly ask the reviewer to account for the fact that we already experimented with models of two distinct scales, $120$M and $1.1$B, and that we achieved similar results in both scales. It is possible, albeit unlikely, that this trend would change on $7$B, $70$B, $700$B or $7$T parameter models.
>
> We need to question ourselves if compute availability is a desirable gate-keeping mechanism for accessing ML conferences and sharing new ideas.
> Why not betting on *novel creative approaches* with substantial impact on our understanding of models?
> Should research now be conducted only by Google, Meta, or any other big-tech lab capable of training very large models? Haven't we learned anything from that kind of behavior in peer-reviewing?
>
>
> # Weakness 2. Narrow Evaluation.
> We argue that the reviewer was vague in this comment as they did not list the ``broader benchmarks and domain" corpora that would benefit our paper.
> We kindly remind the reviewer that we already performed both in-context evaluation (perplexity in two datasets) and context extrapolation (passkey retrieval and Ruler).
>
> Nevertheless, to improve our evaluation, we are running instruction tuning in our $1$B models and will provide results on arc-easy, arc-challenge, MMLU, and LongBenchV2.
> We will be posting a new comment next week with these new results, hoping it properly addresses the reviewers question.
>
>
> # Weakness 3. Training overhead for ultra-long fine-tuning.
> We have implemented FlexAttention [1] in our code base and will re-run inference time analysis that we show in the paper. Next week we will be posting the results.
>
> However, our PE method does not demand "ultra-long" fine-tuning. We argue that such fine-tuning regimes are evidence of the unfulfilled promise of PE to allow context length extrapolation. We argue that this is not a weakness, but rather a strength of our method.
>
>
> # Q1 - Optimizing Z
> Since the best results were achieved by using Scalable Softmax, the scaling factor is a learnable parameter.
> Our experiments show that context extrapolation is improved when using SSMax (see appendix F4).
>
>
> # Q2 - Distinct training regimes
> We do that in Appendix F3, where we perform ablation in training context size. We evaluate $512$, $1024$ and $2048$ tokens of training context size.
>
> # Q3 - Large Models Overfit
> This is an interesting question, but we need to keep in mind that all PE to date overfit in the training context length. However BAM can easily extrapolate by performing a very light fine-tuning ($256$ steps in $1024$ context length) whereas other models still remain stagnated at the pretraining context length.
>
>
> # Q4 - Per layer $\beta$
>
> Our trainable parameters are already different per layer. We perform an ablation with the trainable parameters of our model (see appendix F2) and we see that the best context extrapolation is achieved when training both $\theta_\beta$ and $\theta_\alpha$. We maintain this configuration throughout the paper.

---

> > ### Comment · Reviewer_b3U5 · 2025-11-25
> > **Re: Model scale**
> >
> > After reading the authors' rebuttal, I appreciate the clarifications regarding computational constraints.
> >
> > **On compute accessibility**: I want to clarify that my request for larger-scale validation concerns **epistemic confidence**, not gatekeeping. As an anonymous reviewer, I cannot know your resources, and I genuinely empathize with compute limitations. Novel ideas should not require big-tech budgets. However, many techniques that succeed at 100M-1B scale encounter unexpected challenges at 7B+ due to optimization dynamics or emergent behaviors. Evidence of scalability—even through lightweight proxies—substantially increases **credibility and adoption potential**. This concern is methodological, not about access barriers.
> >
> > That said, I've tailored my suggestions below to be **feasible with existing checkpoints**, requiring no additional large-scale training.

---

> > ### Comment · Reviewer_b3U5 · 2025-11-25
> > **Re: Suggestions**
> >
> > ### 1. Scaling Behavior (Weakness 1)
> >
> > The authors' compute limitations are understood. Instead of training larger models, it would be useful to present a **parameter trend analysis** across the three model scales (120M, 430M, and 1.1B) showing how the learned $\theta_\beta$ and $\theta_\alpha$ distributions evolve. Demonstrating consistent trends would support claims about generalization.
> >
> > Additionally, a brief **theoretical analysis** could clarify that BAM introduces only O(3 × heads × layers) additional parameters, which remain negligible even at 7B+ scale, providing a principled argument for why the observed trends should transfer to larger models.
> >
> > (If a 2–3B checkpoint becomes available, even limited evaluation would be informative, although the existing three scales already provide meaningful evidence for assessing scaling behavior.)
> >
> > ---
> >
> > ### 2. Evaluation Breadth (Weakness 2)
> >
> > The commitment to include instruction-tuned results is appreciated. A minimal representative subset would suffice to address the evaluation narrowness concern:
> >
> > - ARC-easy and ARC-challenge
> > - 5–10 representative MMLU subjects (rather than all 57)
> > - 2–3 LongBench tasks (e.g., passage retrieval, multi-document QA)
> >
> > This would demonstrate generalization across task types without imposing excessive computational burden.
> >
> > ---
> >
> > ### 3. Training Overhead Analysis (Weakness 3)
> >
> > Table 6 provides useful inference timing results, but the training overhead could be further quantified. Reporting the following would give a more complete picture:
> >
> > - Per-step wall-clock time (BAM vs. baselines)
> > - Peak GPU memory consumption at 512/1024 context lengths
> > - Steps to reach target perplexity (convergence comparison)
> >
> > These metrics can likely be extracted from existing training logs. I acknowledge that your method's minimal fine-tuning requirement (256 steps at 1024 context) is indeed a strength: the requested metrics would quantify this advantage numerically.
> >
> > ---
> >
> > ### 4. Q1–Q4 Clarifications
> >
> > **Q1** (Normalization factor Z): A plot showing how the learned SSMax scale parameter correlates with context length would clarify the optimization of Z.
> >
> > **Q2** (Training context regimes): Appendix F.3 contains relevant ablations. A summary figure depicting the relationship between training context and extrapolation ceiling would improve clarity.
> >
> > **Q3** (Why larger models overfit more): Lightweight diagnostics on existing checkpoints could provide insight:
> >
> > - Attention entropy across layers
> > - Distributions of learned $\beta$ and $\alpha$ (are they more peaked/local in larger models?)
> > - Loss landscape near the training context boundary
> >
> > **Q4** (Per-layer strategies): To clarify: my question explored whether freezing early layers or using layer-specific $\beta$ initialization schedules might offer additional benefits. A 50 to 100 step pilot would be informative but is not essential.
> >
> > ---
> >
> > ### Overall Assessment
> >
> > The conceptual contribution of the paper remains significant. The suggested additions are feasible within a one-week timeframe using existing checkpoints and logs, and their inclusion would substantially strengthen the empirical evidence.
> >
> > I am willing to raise my score if these revisions are incorporated. Items 2 and 3 are particularly feasible as they require minimal additional computation, primarily analysis of existing results and evaluation on standard benchmarks.
> >
> > The paper presents a valuable theoretical contribution, and these empirical additions would strengthen its case for acceptance.

---

> ### Author Response · Authors · 2025-11-28
>
> We thank the reviewer for acknowledging our compute limitations. We agree that addressing all the questions and points raised in their comment would drastically improve our paper.
> Here is our new results addressing *all* the points mentioned by the reviewer.
>
> # Weakness 1
>
> We introduced Figure 17 and Figure 18, showing how $\theta_\alpha$ and $\theta_\beta$ behave across model sizes ($120$M, $400$M and $1.1$B) and across training context lengths ($512$, $1024$, and $2048$). Curiously, we clearly see three clusters of $(\theta_\alpha,\theta_\beta)$ pairs both across context lengths and model sizes: the first corresponds to negative $\theta_\beta$, which we have already shown in Figure 4a that works as retrieval heads, where attention weights are higher for tokens further away from the query position; the second corresponds to positive $\theta_\beta$, which works similarly to an ALiBi PE (Figure 4b); the third cluster is less interpretable, with negative exponents smaller than $-0.6$. We conjecture that those are more aggressive retrieval heads, but we highlight that this cluster has fewer pairs than the other two.
>
> The stability on both $\theta_\alpha$ and $\theta_\beta$ across distinct model scales and context lengths shows that the proposed Bayesian interpretation of PE has sound empirical evidence.
>
> Regarding the theoretical analysis, memory scaling of our method is linear in attention heads and layers, hence $\mathcal{O}(h\times l)$.
>
> # Weakness 2
> Here are the full results of LongBenchv2, MMLU, ARC-Easy, and Arc-Challenge.
> We see that GGD-BAM outperforms RoPE in all evaluated benchmarks.
>
>
> | | MMLU | ARC-Easy | **ARC-Challenge** |
> | :--- | :---: | :---: | :---: |
> | BAM SSMax | **0.3716** | **0.5770** | **0.4132** |
> | RoPE SSMax | 0.3573 | 0.5715 | 0.4123 |
> ----
>
>
> | | BAM SSMax | RoPE SSMax |
> | :--- | ---: | ---: |
> | Code Repository Understanding | **41.7** | 25.0 |
> | Long In-context Learning | **36.4** | 30.3 |
> | Long-dialogue History Understanding | 35.0 | 35.0 |
> | Multi-Document QA | **26.5** | 25.3 |
> | Single-Document QA | **26.5** | 18.8 |
> | Overall | **28.6** | 24.2 |
> ----
>
> # Weakness 3
> We implemented FlexAttention [1] both in RoPE and GGD-BAM. We updated Table 6 to show throughput and memory consumption on longer contexts for both methods. We notice that, indeed, GGD-BAM introduces some memory overhead, but it is about $1\%$ while giving better results both in in-context and context extrapolation. Therefore, we believe this small overhead to be justifiable.
>
> # Question 1
> From the SSMax paper [2], we have:
>
> $$\sigma(z_i)=\frac{\exp\big({s\ln(n)z_i}\big)}{\sum _{j=1}^n\exp\big(s\ln(n)z_j\big)}$$
> where $n$ is the sequence length size and $s$ is a learnable parameter. By performing a similar procedure of the proof of our Theorem 1, we see that $s\ln(n)$ becomes part of the normalizing scalar $Z$.
> We added Figure 14 in the paper showing a histogram of the learned $s$. All model sizes have a similar skewed distribution with values close to zero and a tail to the right.
>
>
> # Question 2
> We think Figure 12 provides a good summary of the training context against extrapolation ceiling. We improved our explanation to highlight that training with context length $2048$ improves generalization to $90$\% accuracy up to the evaluated length of $512$k, whereas training with context length $512$ makes the model achieve only $40$\% accuracy at $512$k.

---

> ### Author Response · Authors · 2025-11-28
>
> # Question 3
>
> The new Figures 17 and 18 show that both bigger and smaller models have the same trend regarding $\theta_\alpha$ and $\theta_\beta$.
> We identify three linearly-separable clusters of parameters: the first cluster has $\theta_\beta>0$, where each attention head works similarly to a Laplace distribution (ALiBi); the second cluster has  $-0.6<\theta_\beta<0$, which works as a retrieval head; the third cluster has fewer instances than the other two, with $\theta_\beta<-0.6$. We conjecture that this cluster works as a more aggressive retrieval head.
>
> We used these three clusters to tabulate attention entropy as suggested by the reviewer:
>
> |   Layer | $\theta_\beta<-0.6$   | $-0.6<\theta_\beta<0$   | $\theta_\beta>0$   |
> |--------:|:--------------|:--------------|:---------------|
> |       1 | -             | 1.73 ± 1.24   | 1.39 ± 1.06    |
> |       2 | -             | 1.45 ± 0.85   | 1.03 ± 0.86    |
> |       3 | 1.16 ± 0.79   | -             | 0.82 ± 0.84    |
> |       4 | -             | -             | 0.69 ± 0.75    |
> |       5 | -             | -             | 0.61 ± 0.65    |
> |       6 | 0.72 ± 0.56   | 0.61 ± 0.57   | 0.86 ± 0.69    |
> |       7 | 0.85 ± 0.60   | 0.38 ± 0.44   | 0.87 ± 0.70    |
> |       8 | 0.38 ± 0.41   | 0.55 ± 0.50   | 0.73 ± 0.64    |
> |       9 | -             | 0.36 ± 0.45   | 0.60 ± 0.62    |
> |      10 | 0.52 ± 0.51   | 0.71 ± 0.57   | 0.57 ± 0.54    |
> |      11 | 0.82 ± 0.70   | 0.90 ± 0.73   | 0.73 ± 0.70    |
> |      12 | 1.05 ± 0.80   | 0.91 ± 0.74   | 0.80 ± 0.71    |
> |      13 | 0.99 ± 0.79   | 1.10 ± 0.85   | 0.92 ± 0.74    |
> |      14 | 1.03 ± 0.84   | 1.00 ± 0.80   | 1.08 ± 0.79    |
> |      15 | 0.84 ± 0.69   | 0.88 ± 0.71   | 0.87 ± 0.72    |
> |      16 | -             | 0.92 ± 0.68   | 0.97 ± 0.73    |
> ----
>
> We did not find any meaningful patterns in this particular analysis.
>
> We also added Figure 15 to the paper to show how the loss function behaves near the context boundary. As we can see there, after context extension the loss values become more stable and do not increase in longer contexts when compared to the model without context extension.
>
>
> ## References
> 1. Dong, J., Feng, B., Guessous, D., Liang, Y., & He, H. (2024). Flex attention: A programming model for generating optimized attention kernels. arXiv preprint arXiv:2412.05496.
> 2. Nakanishi, K. M. (2025). Scalable-softmax is superior for attention. arXiv preprint arXiv:2501.19399.

---

### Official Review · Reviewer_R1bm · 2025-10-29

**Soundness:** 3
**Presentation:** 2
**Contribution:** 3
**Rating:** 2
**Confidence:** 3

**Summary:**

This paper introduces a new positional encoding design, BAM, which is a variant and improvement of the existing ALiBi.
The authors reformulate current positional encodings from a Bayesian perspective and provide a clear motivation for their proposed modification. Experiments in long-context perplexity and in-context copying tasks (PassKey Retrieval and NIAH) demonstrate BAM's effectiveness in length extrapolation for language models with up to 1B parameters.

**Strengths:**

- The perspective on improving ALiBi is both interesting and well-motivated.

- The authors provide a theoretical analysis that explains their motivation and supports the proposed approach.

- The authors conduct extensive experiments to demonstrate the effectiveness of their method, providing compelling evidence for its validity.

**Weaknesses:**

- Some experimental results are not convincing enough to support the authors’ contributions (See Questions).

- The presentation of this paper could be improved.
The section on theoretical proofs somewhat detracts from the main focus of the work.
While naming the new PE "Bayesian Attention Mechanism" is justifiable, it make sense but feels somewhat forced.
Several important experimental conclusions are scattered throughout the Appendix, which makes it challenging for reviewers to follow.

**Questions:**

---
Q1: Can the authors include additonal baselines to better demonstrate the effectiveness of BAM methods?

Some relevant methods are missing and should be discussed in the experiments. For example, the RoPE variants (PI, NTK, YaRN, etc), which build on RoPE, should be included .
They are the unique benefits of RoPE design, enabling length extraploation either at little to no cost.

Moreover, discussing the latest design in relative positional encodings [1] would further strengthen the case for BAM's effectiveness.

[1] HoPE: A Novel Positional Encoding Without Long-Term Decay for
Enhanced Context Awareness and Extrapolation.

---
Q2: Can the authors continual pre-train these models with more tokens?

We appreciate the authors's efforts in conducting these experiments.
However, the training corpus used for the pre-training of these toy models remains limited.
Given this, conclusions drawn from such a limited number of pre-training tokens are not entirely convincing.
It is well known that ALiBi can accelerate convergence during training.
So, will BAM still exhibit its advantages when pre-training with more tokens (at least 100B tokens)?
Hope the authors can clarify that.

---
Q3: More comprehensive evaluation of the long-context abilities.

The existing experiments (PassKey and NIAH) provide some preliminary evidence of BAM's in-context copying ability for long contexts.
However, BAM's other abilities in long-context is still underexplored, such as many-shot ICL, integration, and reasoning?

Building on the resolution of Q1 & Q2, could the authors test BAM on real-word tasks of long-context benchmarks (e.g., LongBench and L-Eval)?

---
Q4: What's the generative abilities of these PEs with normal lengths.

We acknowledge the authors' experiment in Appendix D.1, where RoPE shows lower perplexity than BAM on Wikipedia.
We agree with the authors that perplexity alone does not accurately reflect these models' ability.
So, what is the performance of these models at different scales, using different PEs, on language modeling tasks such as ARC, Hellaswag, PiQA, etc.?
Could the length-extrapolation ability of BAM emerge at the cost of its performance on shorter texts?

---

> ### Author Response · Authors · 2025-11-21
>
> We thank the reviewer for engaging in our paper and providing constructive feedback.
> However, we want to make it clear that the main (and most important) contribution of our study is a *theoretical framework* that frames attention within a Bayesian perspective. We consider this new formulation to be an exciting discovery with significant implications.
>
> To demonstrate the importance and usefulness of the proposed theoretical framework, we showed how easy it is to design a completely novel PE strategy as an instance of the proposed framework, and targeted on context extrapolation.
> Our empirical results, secondary to our main contribution, show that the designed method is *by far* the best PE strategy in the *zero-shot context extrapolation scenario*.
> Regarding the weakness pointed by the author that the section on theoretical proofs somewhat detracts from the main focus, we kindly disagree, as we see the theoretical framework as our main contribution and the core of our work.
> We highlight that both NoPE and ALiBi had only ad-hoc explanations on their inner works until our paper shed light on it.
> The authors of ALiBi arrived exactly in a Laplacian distribution through an entirely empirical and trial-and-error fashion.
>
> We kindly ask the reviewer to read the README file on ALiBi's paper. In the FAQ section you will find as follows:
> "(...) Why do you think ALiBi works? I'm not quite sure (...)"
>
> Here is the repository for ALiBi: https://github.com/ofirpress/attention\_with\_linear\_biases
>
> There is a lot of evidence about the ad-hoc nature of previous PE. In ALiBi's  README repository, you will find some:
> "(...) during the development of ALiBi I tried making the bias growth function exponential and that performed worse than linear. I tinkered around with this a bit, trying other growth functions, but linear worked best (...)"
>
> So our contributions provide a clear and concise explanation regarding how PE methods work and how they can be interpreted in a probabilistic manner. This framework has the potential to change how we see and deal with PE altogether.
>
> We now address the four questions pointed by the reviewer.
>
> # Regarding Q1 - More Baselines
>
> We appreciate the suggestion to expand our comparative analysis. However, our current evaluation already encompasses $12$ distinct baselines, comparing GGD-BAM against RoPE, RoPE Local, ALiBi, NoPE, and Sinusoidal PE, both with and without Scalable Softmax.
> Crucially, our implementation aligns with the Scalable Softmax framework [1], which inherently utilizes non-linear position interpolation analogous to NTK-Aware [6] and Code Llama [2]; thus, the benefits of these designs are effectively represented in our experiments.
> We excluded standard Position Interpolation [3] as it is a linear method that lacks the zero-shot extrapolation capabilities central to our analysis.
> Regarding the other suggested variants: while YaRN [4] introduces non-linearity to the exponentially-decaying frequencies, it increases implementation complexity without yielding significant improvements in context extrapolation for our specific comparisons.
> Similarly, the paper introducing HoPE [5] indicates performance decay in Needle-In-A-Haystack tasks during sequence extrapolation.
> When looking to the published version of HoPE, we can see that it fails to generalize even inside de $8$k trained context (as the authors show in their NIAH evaluation).
> Therefore, we focused our comparison on foundational encoding methods rather than an exhaustive enumeration of every RoPE interpolation variant.
>
> # Regarding Q2 - Training with more tokens
> We are not entirely sure how training for more tokens would improve our theoretical framework, or how it would change the superiority of our proposed PE in comparison to others.
> We followed Chinchilla scaling laws to find the proper number of tokens to train our models. We are aware that models perform better after training with more data, but there are diminishing returns.
> Our $120$M models, for instance, were trained for more than $4\times$ the minimum number of tokens recommended by chinchilla scaling laws.
> Note that the general trend tends to be the same; ALiBi, for instance, still works as local attention, even when trained for many more tokens.
>
> Although extra-training can lower the loss function and perplexity, little-to-no effect is expected in long context. This is evident as commercial LLMs such as LLAMA have an additional training phase of context extension (which is basically a pretty expensive fine tuning with context-size around $100$k tokens, in a LongRoPE fashion as described by META Team in their blog post).

---

> > ### Author Response · Authors · 2025-11-21
> > **continuation**
> >
> > # Regarding Q3 - More comprehensive evaluation of the long-context abilities
> >
> > Although passkey retrieval and NIAH are good synthetic tasks that isolate the model capability to use information in longer context, we agree with the reviewer that additional experiments in other tasks can help the reader to be more confident on the superiority of BAM's instances in longer context.
> >
> > We are performing instruction tuning in our $1.1$B parameter models and running the ARC-easy, ARC-challenge, and MMLU benchmarks to evaluate in-context capabilities.
> > To better address context extrapolation, we will provide results also in LongBenchV2 next week, in an additional comment.
> >
> > # Regarding Q4 - Generative abilities with normal lengths
> > We kindly remind the reviewer that we have in-context perplexity evaluation on two distinct datasets.
> > Even though it is not the focus of our study to improve in-context model performance, we agree with the reviewer that such benchmarks are standard in LLM evaluation.
> > To address this gap, we are running instruction tuning in our $1$B models and will provide results on arc-easy, arc-challenge, MMLU, and LongBenchV2.
> > We will be posting a new comment next week with these new results, hoping it properly addresses the reviewers question.
> >
> > 1. NAKANISHI, Ken M. Scalable-softmax is superior for attention. arXiv preprint arXiv:2501.19399, 2025.
> > 2. ROZIÈRE, Baptiste et al. Code Llama: Open foundation models for code. arXiv preprint arXiv:2308.12950, 2023.
> > 3. CHEN, Shouyuan et al. Extending context window of large language models via positional interpolation. arXiv preprint arXiv:2306.15595, 2023.
> > 4. PENG, Bowen et al. YaRN: Efficient context window extension of large language models. arXiv preprint arXiv:2309.00071, 2023.
> > 5. LUO, Yuhang et al. HoPE: A Novel Positional Encoding Without Long-Term Decay for Enhanced Context Awareness and Extrapolation. arXiv preprint arXiv:2410.21216, 2024.
> > 6. BLOC97. NTK-Aware scaled RoPE allows LLaMA models to have extended (8k+) context size without any fine-tuning and with minimal perplexity degradation. Reddit Post, 2023.

---

> ### Author Response · Authors · 2025-11-28
>
> We have posted novel results and additional experiments that we believe directly strengthen the paper and address this reviewer's key concerns.
>
> Here are the novel results on downstream tasks and on long context regarding MMLU, ARC-easy, ARC-challenge, and LongBenchV2.
> We see that GGD-BAM outperforms RoPE in all evaluated benchmarks.
>
>
> | | MMLU | ARC-Easy | **ARC-Challenge** |
> | :--- | :---: | :---: | :---: |
> | BAM SSMax | **0.3716** | **0.5770** | **0.4132** |
> | RoPE SSMax | 0.3573 | 0.5715 | 0.4123 |
> ----
>
>
> | | BAM SSMax | RoPE SSMax |
> | :--- | ---: | ---: |
> | Code Repository Understanding | **41.7** | 25.0 |
> | Long In-context Learning | **36.4** | 30.3 |
> | Long-dialogue History Understanding | 35.0 | 35.0 |
> | Multi-Document QA | **26.5** | 25.3 |
> | Single-Document QA | **26.5** | 18.8 |
> | Overall | **28.6** | 24.2 |
> ----
>
> We have updated the submitted manuscript with those novel results.

---

### Official Review · Reviewer_giCK · 2025-10-30

**Soundness:** 3
**Presentation:** 3
**Contribution:** 3
**Rating:** 6
**Confidence:** 1

**Summary:**

This paper proposes Bayesian Attention Mechanism (BAM), a probabilistic formulation of self-attention where positional encoding (PE) is treated as an explicit prior over token positions.

**Strengths:**

Disclaimer: I don't know the field of positional encoding. I've alerted AC for my lack of knowledge in the domain.

The method can enable retrieval for context length far beyond training context length with small number of parameters added, which seems nice to me.

**Weaknesses:**

Disclaimer: I don't know the field of positional encoding. I've alerted AC for my lack of knowledge in the domain.

I'm unsure about the experimental analysis section of the paper. I don't know if the benchmarks used in the paper includes most of the popular ones in the field of positional encoding. I don't know if the paper uses enough baselines to compare their method with.

**Questions:**

N/A

---

> ### Author Response · Authors · 2025-11-19
> **Answer to Reviewer giCK**
>
> We thank the reviewer for reading our paper and for trying its best to provide feedback while recognizing they are not an expert in PE.
> To help the reviewer be more confident in our work, we assembled a table showing previous work on PE and how they were evaluated:
>
> | Method Name | Category | Year | Conference | Multi-Model Size | Perplexity | Perplexity (Multi-Dataset) | NIAH/Passkey Retrieval | Complete LongBench | Complete Ruler |
> | :--- | :--- | :---: | :--- | :---: | :---: | :---: | :---: | :---: | :---: |
> | ALiBi [1] | PE | 2022 | ICLR | Yes | Yes | Yes | No | NA | NA |
> | KERPLE [2] | PE | 2022 | NeurIPS | No | Yes | Yes | No | NA | NA |
> | Sandwich [3] | PE | 2023 | ACL | No | Yes | Yes | Yes | No | NA |
> | XPos [4] | PE | 2023 | ACL | No | Yes | Yes | No | No | NA |
> | NoPE [5] | PE | 2023 | NeurIPS | No | Yes | Yes | No | No | NA |
> | YaRN [6] | Context Extension | 2023 | ICLR | Yes | Yes | No | Yes | No | NA |
> | BiPE [7] | PE | 2024 | ICML | Yes | Yes | Yes | No | No | No |
> | DAPE [8] | PE | 2024 | NeurIPS | Yes | Yes | Yes | No | No | No |
> | POSE [9] | Context Extension | 2024 | ICLR | Yes | Yes | Yes | Yes | No | No |
> | LongRoPE [10] | Context Extension | 2024 | ICML | No | Yes | Yes | Yes | No | No |
> | Wavelet [11] | PE | 2025 | ICLR | No | Yes | No | No | Yes | No |
> | TAPE [12] | PE | 2025 | ICML | No | Yes | Yes | Yes | No | No |
> | HoPE [13] | PE | 2025 | ACL | Yes | Yes | No | Yes | No | No |
> | BAM (Ours) | PE | 2025 | - | Yes | Yes | Yes | Yes | No | No |
>
> This table clearly shows that among papers proposing PE methods, our evaluation of GGD-BAM is one of the most comprehensive to date. We test across multiple datasets and model sizes, and incorporate a wider array of evaluations, including perplexity, passkey retrieval, and Needle-In-A-Haystack (NIAH), than any *direct peer* in our category.
>
> Considering the comments from other reviewers, we are working on improving our in-context evaluation with the following benchmarks: ARC-easy, ARC-challenge, and MMLU. We are also working on releasing results on LongBenchV2.
> We will post an additional comment next week with these new results.
>
> Please let us know if there is anymore evidence we can present to increase your confidence in the soundness and novelty of our work.
>
>  1. Press, O., Smith, N. A., \& Lewis, M. (2021). Train short, test long: Attention with linear biases enables input length extrapolation. arXiv preprint arXiv:2108.12409.
>  2. Chi, T. C., Fan, T. H., Ramadge, P. J., \& Rudnicky, A. (2022). Kerple: Kernelized relative positional embedding for length extrapolation. Advances in Neural Information Processing Systems, 35, 8386-8399.
>  3. Chi, T. C., Fan, T. H., Rudnicky, A. I., \& Ramadge, P. J. (2022). Dissecting transformer length extrapolation via the lens of receptive field analysis. arXiv preprint arXiv:2212.10356.
>  4. Yutao Sun, Li Dong, Barun Patra, Shuming Ma, Shaohan Huang, Alon Benhaim, Vishrav Chaudhary, Xia Song, and Furu Wei. 2023. A Length-Extrapolatable Transformer. In Proceedings of the 61st Annual Meeting of the Association for Computational Linguistics (Volume 1: Long Papers), pages 14590–14604, Toronto, Canada. Association for Computational Linguistics.
>  5. Kazemnejad, A., Padhi, I., Natesan Ramamurthy, K., Das, P., \& Reddy, S. (2023). The impact of positional encoding on length generalization in transformers. Advances in Neural Information Processing Systems, 36, 24892-24928.
>  6. Peng, B., Quesnelle, J., Fan, H., \& Shippole, E. (2023). Yarn: Efficient context window extension of large language models. arXiv preprint arXiv:2309.00071.
>  7. He, Z., Feng, G., Luo, S., Yang, K., Wang, L., Xu, J., ... \& He, D. (2024). Two stones hit one bird: Bilevel positional encoding for better length extrapolation. arXiv preprint arXiv:2401.16421.
>  8. Zheng, C., Gao, Y., Shi, H., Huang, M., Li, J., Xiong, J., ... \& Li, Y. (2024). Dape: Data-adaptive positional encoding for length extrapolation. Advances in Neural Information Processing Systems, 37, 26659-26700.
>  9. Zhu, D., Yang, N., Wang, L., Song, Y., Wu, W., Wei, F., \& Li, S. (2023). Pose: Efficient context window extension of llms via positional skip-wise training. arXiv preprint arXiv:2309.10400.
>  10. Ding, Y., Zhang, L. L., Zhang, C., Xu, Y., Shang, N., Xu, J., ... \& Yang, M. (2024). Longrope: Extending llm context window beyond 2 million tokens. arXiv preprint arXiv:2402.13753.
>  11. Oka, Y., Hasegawa, T., Nishida, K., \& Saito, K. (2025). Wavelet-based positional representation for long context. arXiv preprint arXiv:2502.02004.
>  12. Zhu, J., Wang, P., Cai, R., Lee, J. D., Li, P., \& Wang, Z. (2025). Rethinking addressing in language models via contexualized equivariant positional encoding. arXiv preprint arXiv:2501.00712.
>  13. Chen, Y., Lv, A., Luan, J., Wang, B., \& Liu, W. (2024). Hope: A novel positional encoding without long-term decay for enhanced context awareness and extrapolation. arXiv preprint arXiv:2410.21216.

---

### Official Review · Reviewer_xnWw · 2025-11-01

**Soundness:** 2
**Presentation:** 3
**Contribution:** 2
**Rating:** 4
**Confidence:** 3

**Summary:**

This paper introduces the Bayesian Attention Mechanism (BAM), a theoretical framework that reformulates self-attention as a probabilistic model where positional encoding (PE) acts as a prior over token positions. The authors show that existing PEs like NoPE and ALiBi correspond to specific priors (Uniform and Laplace) and propose a new Generalized Gaussian prior (GGD-BAM) to improve long-context extrapolation. They validate the approach empirically on language modeling, passkey retrieval, and the RULER benchmark.

**Strengths:**

1. The paper provides a clear and rigorous probabilistic interpretation of positional encoding in self-attention, unifying multiple existing approaches under a single Bayesian view. The derivations are mathematically consistent and offer new insight into the role of priors in attention mechanisms.
2. Framing positional encoding as a prior elegantly connects methods like NoPE and ALiBi that were previously viewed as unrelated heuristics. This contributes to a more principled understanding of extrapolation in transformers.
3. The visualizations of positional priors and learned β values provide interpretability, showing how certain heads specialize in long-range retrieval.

**Weaknesses:**

1. The main experiments are restricted to models with limited parameters. It remains unclear whether the extrapolation benefits persist at the scale of modern large language models (7B–70B), where positional behaviors often change.
2. The work focuses on retrieval and perplexity but does not assess downstream tasks like long context QA or summarization, where long-context comprehension is critical.

**Questions:**

See above.

---

> ### Author Response · Authors · 2025-11-21
>
> We thank the reviewer for engaging in our paper and providing constructive feedback.
> However, we want to make it clear that the main (and most important) contribution of our study is a *theoretical* framework that frames attention within a Bayesian perspective. We consider this new formulation to be an *exciting discovery* with significant implications.
>
> To demonstrate the importance and usefulness of the proposed theoretical framework, we showed how easy it is to design a *completely novel* PE strategy as an instance of this framework, which is targeted on context extrapolation.
> Our empirical results, secondary to our main contribution, show that the designed method is by far the best PE strategy in the *zero-shot context extrapolation scenario*.
>
> We now address the two weaknesses pointed by the reviewer.
>
> # Regarding Weakness 1 - Model Scale
> The reviewer is right in his observation that our experiments are restricted to models ranging from $120$M to $1.1$B parameters.
> We are a University research lab with limited computational budget (as pointed in our experimental setup section of the paper).
> We have access to 4 NVidia A6000 GPUS with 48GB Vram each.
> In our setup, the training of a $1.1$B parameter model takes almost two entire weeks of exclusive allocation in our servers. So training a single $7$B parameter model is a task that would demand almost $420$ days of our compute (taking into account we would need $7\times$ more parameters and $7\times$ more data, a $49\times$ increase in time is a conservative estimate).
> A $70$B model would not even fit in our available budget.
>
> We kindly ask the reviewer to account for the fact that we already experimented with models of two distinct scales, $120$M and $1.1$B parameters, and we achieved similar results in both scales.
> Our entire validation process clearly holds for Small Language Models (SMLs), but we do not discard the hypothesis that it would also work for LLMs.
> It is possible, albeit unlikely, that this trend could change on $7$B, $70$B, $700$B, or $7$T-parameter models.
>
> We need to question ourselves if compute availability is a desirable gate-keeping mechanism for accessing ML conferences and sharing new ideas.
> Why not betting on *novel creative approaches* with substantial impact on our understanding of models?
> Should research now be conducted only by Google, Meta, or any other big-tech lab capable of training very large models? Haven't we learned anything from that kind of behavior in peer-reviewing?
>
> # Regarding Weakness 2: Downstream tasks
>
> We are performing instruction tuning in our $1.1$B parameter models and will execute the LongBenchV2 benchmark, which encompasses downstream tasks and not only NIAH.
> We will post a follow up to this comment as soon as we have the results.

---

> > ### Comment · Reviewer_xnWw · 2025-11-26
> >
> > Thanks for the response.
> >
> > I think addressing W2 will make the work publishable (I would give it a score of 6).
> > Addressing W1 would broaden the impact and make the work more practical (I would give it a score of 8).

---

> > > ### Author Response · Authors · 2025-11-28
> > >
> > > # Regarding Weakness 1 - Model Scale
> > >
> > > We thank the reviewer for their transparency regarding the scoring criteria. While we appreciate the acknowledgment that addressing the model scale could increase the evaluation to an 8, we must respectfully **reiterate** our stance on the requirements for scientific contribution in this field.
> > >
> > > We have demonstrated that our method consistently extrapolates benefits across distinct orders of magnitude (from $125$M to $1$B parameters). To assume that a trend verified on $1$B parameters would suddenly invert at $7$B is a hypothesis that places the burden of the proof on the authors' budget rather than on the method's mathematical soundness.
> > >
> > > Furthermore, we must push back against the premise that access to industrial-scale compute (required for $7$B--$70$B models) should be a prerequisite for a **strong accept**. Conflating scientific novelty with computational capacity risks establishing an inadvertent gatekeeping mechanism, effectively restricting high-tier research output to a handful of industrial labs. As an academic laboratory committed to the democratization of AI and Green AI principles, we believe that novel architectures should be evaluated on their theoretical merit and empirical consistency within accessible constraints, rather than their immediate deployment at industrial scale.
> > >
> > > Deployment at industrial scale should be treated as an engineering challenge not a requirement for publication.
> > >
> > >
> > > # Regarding Weakness 2: Downstream tasks
> > >
> > > Here are the full results of LongBenchv2, MMLU, ARC-Easy, and Arc-Challenge.
> > > We see that GGD-BAM outperforms RoPE in all evaluated benchmarks.
> > >
> > >
> > > | | MMLU | ARC-Easy | ARC-Challenge |
> > > | :--- | :---: | :---: | :---: |
> > > | BAM SSMax | **0.3716** | **0.5770** | **0.4132** |
> > > | RoPE SSMax | 0.3573 | 0.5715 | 0.4123 |
> > > ----
> > >
> > >
> > > | | BAM SSMax | RoPE SSMax |
> > > | :--- | ---: | ---: |
> > > | Code Repository Understanding | **41.7** | 25.0 |
> > > | Long In-context Learning | **36.4** | 30.3 |
> > > | Long-dialogue History Understanding | 35.0 | 35.0 |
> > > | Multi-Document QA | **26.5** | 25.3 |
> > > | Single-Document QA | **26.5** | 18.8 |
> > > | Overall | **28.6** | 24.2 |
> > > ----
> > >
> > > We have updated the submitted manuscript with those novel results.

---

### Author Response · Authors · 2025-12-01
**Final Remarks**

Dear Area Chair,


In our paper, we introduce the Bayesian Attention Mechanism (BAM), a theoretical framework that reframes self-attention as an expectation of values computed under a joint probabilistic model of *content* and *position* of tokens. Within BAM, Positional Encoding strategies naturally emerge as a prior distribution over token positions, clarifying the theoretical basis of existing techniques. Through four different theorems and four lemmas, we mathematically demonstrate, among other things, how NoPE and ALiBi correspond directly to Uniform and Laplace positional priors, respectively. We highlight that both NoPE and ALiBi had only ad-hoc explanations on their original papers until our work shed light on it. The authors of ALiBi arrived exactly at a Laplacian distribution through an entirely empirical and trial-and-error approach. For instance, within the README file from ALiBi's github repository, in the FAQ section, you will find the following Q\&A: "Q:(...) Why do you think ALiBi works? A: I'm not quite sure (...)".

Leveraging this robust theoretical foundation, we show how easy it is to propose new PE strategies with known behaviors. In particular, we propose a new strategy based on a Generalized Gaussian prior, whose explicit mathematical properties allow for long-context retrieval/extrapolation.
Such proposed approach introduces fewer than $1,000$ additional parameters yet delivers substantially improved extrapolation performance, empirically shown when compared to 16 different PE strategies in retrieval-based tasks and traditional perplexity evaluation. BAM serves both as a unified theoretical framework for analyzing PE schemes and as a practical method for designing strategies capable of long-context attention.

Our paper was substantially strengthened during the rebuttal period (see all extra material included in the manuscript colored in red). We addressed **all points** raised by the reviewers. None of them raised concerns regarding our theoretical findings and empirical results - their suggestions were mainly about adding more empirical evidence through the inclusion of new benchmarks.

Specifically, Reviewer b3U525 provided a constructive path towards acceptance of our work, which we diligently followed.
A summary of what we did during rebuttal time:
1) evaluation on MMLU, ARC-easy, and ARC-challenge benchmarks for downstream performance evaluation - our results outperformed RoPE;
2) evaluation on LongBenchV2 benchmark for downstream performance on long context - our results again outperformed RoPE, here by a wide margin;
3) an empirical investigation of $\theta_\beta$ and $\theta_\alpha$, showing that we have three distinct types of probability distributions: two specialized in retrieval and one in local attention.

Finally, we wish to highlight the **only remaining weakness not directly addressed by us** during the rebuttal period: lack of results in **very** large models. Reviewer xnWw explicitly stated they would raise their score to an 8 **only** if we implemented our method within $7$B–$70$B parameter models. We respectfully argue that conditioning an acceptance score to industrial-scale experimental analysis acts as an unintentional gatekeeping mechanism for academic research. We have demonstrated that our method’s benefits extrapolate consistently from $120$M to $1$B parameters. To assume that a trend verified on
$1$B parameters would suddenly invert at $7B$ is a hypothesis that places the burden of the proof on the authors' budget rather than on the method's mathematical soundness. Conflating scientific novelty with computational capacity effectively restricts high-tier research output to a handful of industrial labs. As an academic laboratory committed to the democratization of AI and Green AI principles, we believe that novel ideas should be evaluated on their theoretical merit and empirical consistency within accessible constraints, rather than their immediate deployment at industrial scale. Deployment at industrial scale should be treated as an engineering challenge, not as a requirement for publishing exciting findings.

We again highlight the fact that the main (and most important) contribution of our study is a **theoretical framework** that frames attention within a Bayesian perspective. We consider this new formulation to be an **exciting discovery** with significant implications. Our empirical results, secondary to our main contribution, show that the designed method is the state-of-the-art PE strategy in the zero-shot context extrapolation scenario, being able to retrieve information at 500$\times$ the trained context whereas previous methods struggle in 2-3$\times$. This was largely corroborated by the results of several benchmarks we included during the rebuttal period.

---

### Meta-Review · Area_Chair_VLJm · 2026-01-03

**Summary:**

This paper proposes Bayesian Attention Mechanism (BAM), a probabilistic framework of positional encoding as a prior in attention, and introduces a generalized Gaussian prior variant aimed at improving long-context extrapolation. Reviewers generally found the core idea and theoretical perspective interesting, and the reported long-context retrieval results promising. The main concerns are that the empirical evidence is still not strong enough, especially given (i) limited scale (experiments stop at ~1.1B), and (ii) incomplete coverage of standard baselines and evaluations expected for positional encoding/long-context claims. Even with added rebuttal results, the paper remains insufficiently validated against the strongest and most relevant alternatives and settings. However, I think this paper could provide some interesting insights on positional encoding for the community, so I recommend a borderline accept for this paper.

**Reviewer Concerns:**

### Concerns addressed in the rebuttal: ###

- The authors added results on LongBenchV2 and common downstream benchmarks (e.g., MMLU/ARC), which directly addresses the repeated request to move beyond synthetic retrieval-only evidence.

- The authors provided additional context and positioning of their evaluation relative to prior PE papers, which helps but does not fully solve the concerns of missing comparisons.

### Concerns still outstanding: ###

- Scale generalization to modern LLM regimes (>7B): Multiple reviewers mentioned that positional behaviors can change with scale. The rebuttal does not provide large-model evidence.

- One reviewer requested stronger coverage of RoPE variants / recent methods and more direct apples-to-apples baselines. The rebuttal argues for why some variants are less central, but does not convincingly close the gap with comprehensive comparisons.

- There were concerns that conclusions may depend on relatively limited pretraining tokens and "toy" training conditions. The rebuttal does not add strong new evidence.

- While LongBenchV2 helps, concerns remain about broader long-context behaviors (reasoning, many-shot ICL, integration-style tasks) and about possible trade-offs at normal context lengths.

**Reviewer Scores:**

- Reviewer R1bm (original 2) -> 4: Added downstream results and additional discussion could raise confidence somewhat, but missing key baselines/scale evidence likely keeps them below the bar.

- Reviewer xnWw (original 4) -> 6: The reviewer explicitly indicated that addressing the downstream-task weakness would make the work publishable. The new LongBenchV2/MMLU/ARC results likely achieve that.

- Reviewer giCK (original 6) -> 6: This reviewer stated low confidence and focused on whether the evaluation/baselines are standard. The added benchmarking context and new results help, but the reviewer will likely remain the score.

- Reviewer b3U5 (original 4) -> 6: The new downstream benchmarks directly address the "evaluation is narrow" concern and could move them slightly above threshold, though scale and cost/overhead completeness still limit.

---

### Decision · Program_Chairs · 2026-01-26

Accept (Poster)